# Membrane transporter dimerization driven by differential lipid solvation energetics of dissociated and associated states

Rahul Chadda[1†], Nathan Bernhardt[2†], Elizabeth G Kelley[3], Susana CM Teixeira[3,4], Kacie Griffith[5], Alejandro Gil-Ley[2,5], Tuğba N Öztürk[1], Lauren E Hughes[5], Ana Forsythe[5], Venkatramanan Krishnamani[5], José D Faraldo-Gómez[2]*, Janice L Robertson[1]*

[1]Biochemistry and Molecular Biophysics, Washington University School of Medicine, St. Louis, United States; [2]Theoretical Molecular Biophysics Laboratory, National Heart, Lung and Blood Institute, National Institutes of Health, Bethesda, United States; [3]NIST Center for Neutron Research, National Institute for Standards and Technology, Gaithersburg, United States; [4]Center for Neutron Science, Chemical and Biomolecular Engineering, University of Delaware, Newark, United States; [5]Molecular Physiology and Biophysics, Carver College of Medicine, The University of Iowa, Iowa City, United States

**Abstract** Over two-thirds of integral membrane proteins of known structure assemble into oligomers. Yet, the forces that drive the association of these proteins remain to be delineated, as the lipid bilayer is a solvent environment that is both structurally and chemically complex. In this study, we reveal how the lipid solvent defines the dimerization equilibrium of the CLC-ec1 Cl⁻/H⁺ antiporter. Integrating experimental and computational approaches, we show that monomers associate to avoid a thinned-membrane defect formed by hydrophobic mismatch at their exposed dimerization interfaces. In this defect, lipids are strongly tilted and less densely packed than in the bulk, with a larger degree of entanglement between opposing leaflets and greater water penetration into the bilayer interior. Dimerization restores the membrane to a near-native state and therefore, appears to be driven by the larger free-energy cost of lipid solvation of the dissociated protomers. Supporting this theory, we demonstrate that addition of short-chain lipids strongly shifts the dimerization equilibrium toward the monomeric state, and show that the cause of this effect is that these lipids preferentially solvate the defect. Importantly, we show that this shift requires only minimal quantities of short-chain lipids, with no measurable impact on either the macroscopic physical state of the membrane or the protein's biological function. Based on these observations, we posit that free-energy differentials for local lipid solvation define membrane-protein association equilibria. With this, we argue that preferential lipid solvation is a plausible cellular mechanism for lipid regulation of oligomerization processes, as it can occur at low concentrations and does not require global changes in membrane properties.

*For correspondence:
jose.faraldo@nih.gov (JDF);
janice.robertson@wustl.edu (JLR)

†These authors contributed equally to this work

## Introduction

Lipid bilayers are the most common means of chemical compartmentalization in biology. The bilayer interior, formed by the acyl chains, is a $\approx 30$ Å layer of low-dielectric fluid oil (*Fricke, 1925*) that provides a natural electrostatic barrier for the passage of charged and polar species. This insulating core enables the cell to generate trans-bilayer chemical and electrical potential-energy gradients

**eLife digest** A cell's outer membrane is made of molecules called lipids, which band together to form a flexible thin film, just two molecules thick. This membrane is dotted with proteins that transport materials in to and out of cells. Most of these membrane proteins join with other proteins to form structures known as oligomers. Except, how membrane-bound proteins assemble into oligomers – the physical forces driving these molecules to take shape – remains unclear.

This is partly because the structural, physical and chemical properties of fat-like lipid membranes are radically different to the cell's watery interior. Consequently, the conditions under which membrane oligomers form are distinct from those surrounding proteins inside cells. Membrane proteins are also more difficult to study and characterize than water-soluble proteins inside the cell, and yet many therapeutic drugs such as antibiotics specifically target membrane proteins. Overall, our understanding of how the unique properties of lipid membranes affect the formation of protein structures embedded within, is lacking and warrants further investigation.

Now, Chadda, Bernhardt et al. focused on one membrane protein, known as CLC, which tends to exist in pairs – or dimers. To understand why these proteins form dimers (a process called dimerization) Chadda, Bernhardt et al. first used computer simulations, and then validated the findings in experimental tests.

These complementary approaches demonstrated that the main reason CLC proteins 'dimerize' lies in their interaction with the lipid membrane, and not the attraction of one protein to its partner. When CLC proteins are on their own, they deform the surrounding membrane and create structural defects that put the membrane under strain. But when two CLC proteins join as a dimer, this membrane strain disappears – making dimerization the more stable and energetically favorable option.

Chadda, Bernhardt et al. also showed that with the addition of a few certain lipids, specifically smaller lipids, cell membranes become more tolerant of protein-induced structural changes. This might explain how cells could use various lipids to fine-tune the activity of membrane proteins by controlling how oligomers form. However, the theory needs to be examined further. Altogether, this work has provided fundamental insights into the physical forces shaping membrane-bound proteins, relevant to researchers studying cell biology and pharmacology alike.

that fuel essential metabolic functions. While the macroscopic structure of the lipid bilayer is shared across nearly all species and organelles, their chemical compositions are remarkably diverse. For example, phospholipids can vary in their headgroup moieties, in the length and degree of saturation of the acyl chains, and in the chain-headgroup linkage, i.e. ester vs. ether (*van Meer et al., 2008*). Acyl chains can feature modifications such as branching, or even form covalent bonds across mono-layers, as in tetraether lipids (*Valentine, 2007*). Lipidomics studies indeed show that cellular membranes include hundreds of lipid types (*Brügger, 2014*). It has been proposed that this diversity is in part explained by the 'homeoviscous adaptation' of cells, that is, the need to maintain an appropriate membrane fluidity under a wide variety of environmental conditions (*Sinensky, 1974*). For example, a recent study indicates that under varying dietary fatty acid input, mammalian cells alter the lipid composition of their membranes to regulate this key property (*Levental et al., 2020*). Yet, some of these compensatory chemical changes appear to be excessively redundant. For example, under cold growth temperatures *E. coli* generates unsaturated lipids to increase membrane fluidity, but it also increases production of short-chain lipids (*Marr and Ingraham, 1962*; *Sanders and Mittendorf, 2011*). Do these different chemical strategies target others cellular processes that change coincidentally with variations in fluidity? Is there more to the vast diversity in lipid compositions observed across different types of membranes and conditions, beyond the basic requirement of a fluid lipid bilayer?

One possibility is that this lipid diversity reflects a coupled relationship with the other major constituent of all cellular membranes, namely integral membrane proteins (*Phillips, 2018*). The mechanisms of these proteins are, fundamentally, not unlike those of water-soluble proteins, and entail processes such as molecular recognition, conformational exchange and catalyzed chemistry. For membrane proteins, however, the lipid bilayer provides a distinct reaction environment where lipid

molecules are the primary solvent. In any biological equilibrium reaction, the solvent plays a major role in defining the energetic landscape; it seems therefore logical to hypothesize that the variability in the chemical composition of physiological membranes might reflect adaptive mechanisms of regulation of protein structure and function. A key question is, however, how this kind of regulation can be sufficiently targeted and specific, rather than globally disruptive.

Here, we examine the molecular role of the lipid solvent in a highly prevalent reaction in membrane biology, namely protein oligomerization. Indeed, among membrane-protein classes of known structure, approximately 70% are found as homo- or hetero-oligomers (*Aleksandrova et al., 2020*), compared with about 55% of water-soluble proteins. This comparison is striking because the principal driving force for the formation of protein oligomers in water, that is, the hydrophobic effect (*Tanford, 1978*), cannot be a dominant factor in the membrane, as its interior is largely dehydrated. Membrane protein complexes do bury large non-polar surfaces, bringing many hydrophobic side-chains into close proximity, in the range of van der Waals interactions. Yet, it is unclear whether these kinds of protein-protein contacts are the main drivers for the association of integral membrane proteins (*Cristian et al., 2003*) as these side chains also form numerous, similarly favorable contacts with lipids in the dissociated states. Likewise, it is not evident that interfacial tensions at the protein-lipid boundary are a dominant factor; while the acyl-chain core would favor association to reduce the total area of the protein-lipid interface, the head-group layer has an opposite effect (*Dixit and Lazaridis, 2020*; *Marsh, 2008*).

Nonetheless, it has long been recognized that the complementarity between membrane proteins and their lipid environment is imperfect, resulting in different kinds of perturbations in the structure and dynamics of the bilayer (*Mouritsen and Bloom, 1993*; *Marsh, 2008*). In the context of protein-protein association, local perturbations in membrane thickness are particularly noteworthy; this effect, referred to as 'hydrophobic mismatch', is a key factor in the dimerization equilibrium of helical peptides such as Gramicidin A (*Goforth et al., 2003*; *Andersen and Koeppe, 2007*) and WALP (*Sparr et al., 2005*), and has also been proposed to explain the organization of various rhodopsins and other GPCRs (*Mondal et al., 2013*; *Pearson et al., 1983*; *Soubias et al., 2015*). This type of perturbation results from a suboptimal match between the exposed non-polar surface of a transmembrane protein and the intrinsic thickness of the acyl-chain core of the bilayer, for a given composition. This mismatch typically forces the bilayer to deform, which translates into an energetic penalty; thus, oligomeric states that minimize this penalty are favored, at least in regard to the membrane energetics. Furthermore, because this energetic penalty will depend on the unperturbed bilayer thickness, variations in lipid composition might provide a means for the cell to regulate oligomerization processes (*Andersen and Koeppe, 2007*). However, the potential for this seems limited, as there is an inherent biological drive for cells to maintain the basic biophysical properties of their membrane through homeostatic adaptation (*Leventall et al., 2020*). Yet, in a mixed lipid bilayer that better reflects a biological membrane, it is important to consider that lipids will exhibit differential distributions depending on their energetics. Early lattice models of model membrane systems demonstrated that differential energy terms for lipids around protein surface vs. the bulk would lead to enrichment around proteins, that would in turn provide a lipid-mediated driving force for protein association (*Marcelja, 1976*; *Sperotto and Mouritsen, 1991*; *Sperotto and Mouritsen, 1993*). Thus, we hypothesize that the physiological mechanism of lipid regulated oligomerization equilibrium will involve a molecular mechanism such as this, occurring at low concentrations of regulatory lipids within the membrane, targeting local membrane defects around the protein surface that are introduced by hydrophobic mismatch, and occur in the absence of macroscopic changes in membrane structure.

A key to evaluating the dominant driving forces for membrane protein oligomerization is to develop assays that quantify this kind of equilibria in lipid bilayers with sufficient accuracy and sensitivity to variations in lipid composition. Previously, we established such an assay based on single-molecule fluorescence microscopy, and carried out measurements of the free-energy of dimerization of the *E. coli* Cl⁻/H⁺ antiporter CLC-ec1 (*Figure 1A*) in 2:1 palmityl, oleoyl phosphatidyl-ethanolamine/phosphatidyl-glycerol (2:1 POPE/POPG) lipid bilayers (*Chadda et al., 2016*). These membranes are a synthetic mimic of the *E. coli* polar-lipid content, and consist of C16:0/18:1 acyl-chains, the most commonly found in biological membranes (*Phillips, 2018*). While CLC-ec1 had been known to exist as a homodimer in detergent and membranes (*Maduke et al., 1999*; *Dutzler et al., 2002*), our measurements revealed this complex results from association of two functionally competent

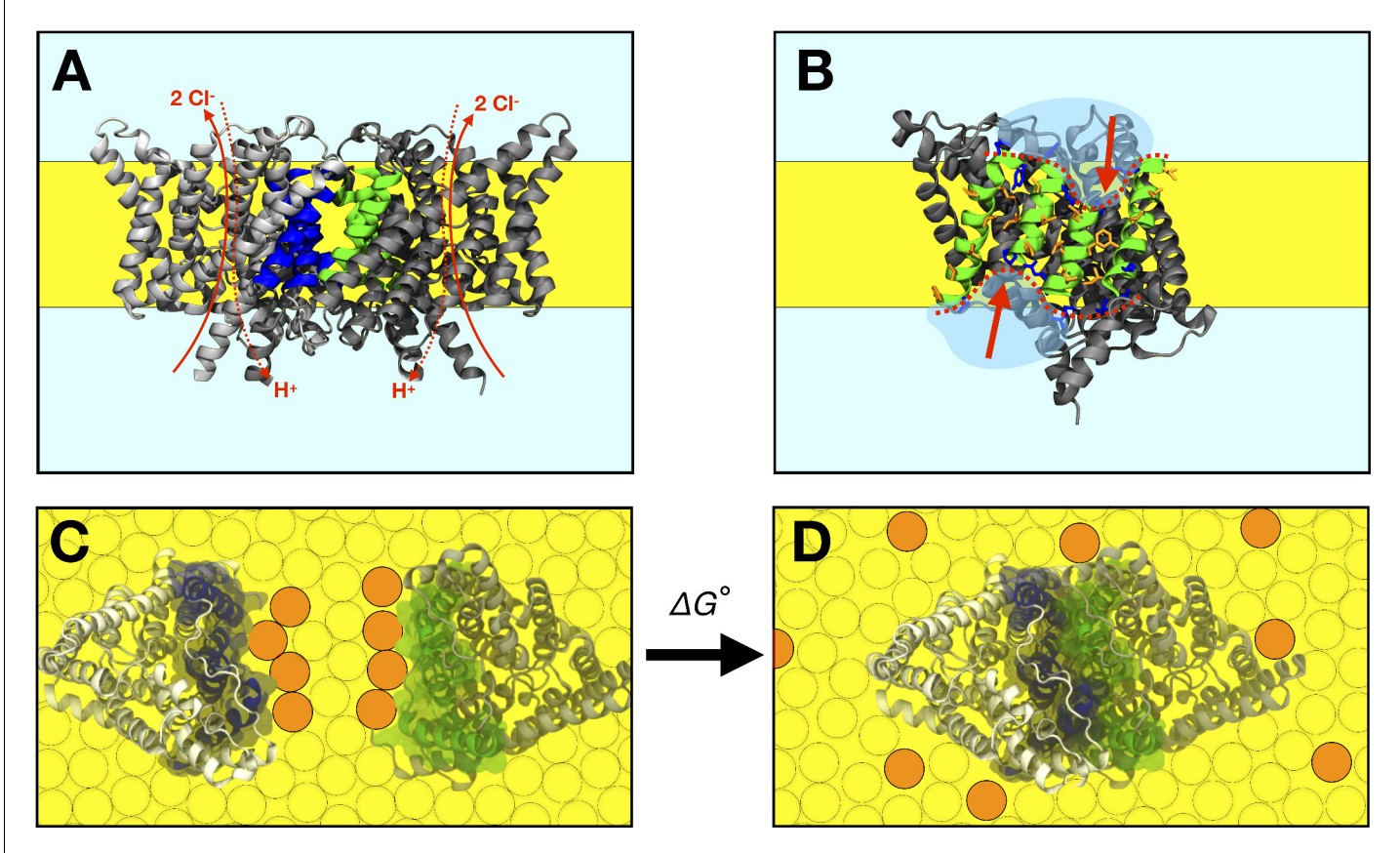

**Figure 1.** The CLC-ec1 dimerization reaction in lipid bilayers. (A) Side view of the CLC-ec1 homodimer in the lipid bilayer. Two subunits are shown in silver and gray, with helices forming the dimerization interface highlighted blue and green. The hydrophobic core of the membrane is depicted in yellow. Approximate pathways for Cl⁻ and H⁺ transport is shown with red arrows. (B) The dimerization interface of the monomer. The four helices forming the interface (Q, P, H, and I) are shown in green, with non-polar side-chains in orange and interfacial polar/charged side-chains in blue. The red dotted line and arrows highlights the shorter H and P helices at the center of the interface. (C,D) Top view of dissociated and associated end-points of the CLC-ec1 dimerization reaction, with the free energy of dimerization defined as $\Delta G^{\circ}$, defined relative to the 1 subunit/lipid mole fraction standard state. Circles represent lipids, with orange circles highlighting an example set of lipids that exchange between the dimerization interface and the bulk upon dimerization.

monomers (*Robertson et al., 2010*), via an interface of about 1200 Å², most of which is inside the membrane (*Figure 1B*). Specifically, the measured equilibrium dimerization free energy for this complex is −10.9 kcal/mole, relative to a standard state of 1 monomer/lipid (*Chadda et al., 2018*). This is a remarkable finding, in that it implies that the population of dissociated monomers at biological protein-expression levels is virtually zero. This interaction is thus reminiscent of obligate water-soluble homomeric complexes, whose association is dominated by the hydrophobic effect (*Bahadur et al., 2003*; *Yan et al., 2008*); by analogy, it is reasonable to infer that the energetics of the lipid solvent might also be key for CLC-ec1 dimerization. Indeed, examination of the dimerization interface shows that the two central helices are much shorter than what is typical in transmembrane segments (*Figure 1B*). In the monomeric state, a significant hydrophobic mismatch might therefore exist between this protein surface and the surrounding membrane (*Figure 1C*), which would be completely eliminated upon dimerization, possibly explaining the remarkable stability of this complex.

CLC-ec1 thus appears to be an excellent system to examine the fundamental questions outlined above. That is, can protein-induced membrane deformations contribute to explain the structure and stability of obligate membrane protein complexes? What is the extent of the changes in membrane lipid composition that are necessary to influence these oligomerization reactions, what is the underlying mechanism, and importantly, are those changes physiologically viable, that is, do they preserve

or impair protein function? To address these questions, we first use molecular dynamics simulations of monomeric and dimeric CLC-ec1 in 2:1 POPE/POPG, to evaluate the lipid bilayer structure in each state. This analysis informs a series of new experimental assays, namely small-angle neutron scattering measurements, Cl⁻ transport assays and single-molecule photobleaching analyses, with which we determine how varying quantities of short-chain C12:0, di-lauryl (DL) lipids alter the dimerization equilibrium as well as the activity of this transporter. To obtain a molecular level interpretation for these new experimental results, we return to molecular dynamics simulations in membrane mixtures that mimic the experimental conditions. These studies lead to a perspective of the dimerization reaction as primarily controlled by the energetics of local lipid solvation of the associated and dissociated states (*Marsh, 1995*; *Marsh, 2008*), and underscore the essential role of molecular-scale heterogeneity of the lipid bilayer in defining membrane protein association equilibria.

## Results

### The CLC-ec1 dimerization interface causes a structural defect in the surrounding membrane

As mentioned, the features of the CLC dimerization interface (*Figure 1B*) suggest that, when exposed in the monomeric state, there might be a hydrophobic mismatch with the surrounding membrane (*Figure 1C*). If so, the energetic cost associated with solvating the monomer could translate into an effective driving force toward dimerization; by burying these 'problematic' interfaces away from the lipids, the system would gain free energy upon association (*Figure 1D*). To begin to validate or refute this hypothesis, we first studied the structure of the lipid bilayer around the CLC-ec1 monomer and dimer, using coarse-grained molecular dynamics (CGMD) simulations, and evaluated whether the exposed dimerization interface indeed appears to cause the membrane to adopt a higher energy state, relative to the other regions of the protein surface. For both monomer and dimer, we used 2:1 POPE/POPG membranes (*Figure 2—figure supplement 1A*), corresponding to the C16:0/18:1 acyl chains used in our previous experimental measurements of the reversible dimerization reaction (*Chadda et al., 2016*).

It is worth noting that any simulation of a non-homogenous membrane necessarily presupposes an initial spatial distribution of the lipid components, which is not only arbitrary but also may not be representative of the equilibrium condition. Prior to examining the structure of these membranes, it is therefore key to ascertain that the simulations are long enough for the two lipid components to mix fully and spatially re-distribute according to the free-energy landscape of the molecular system. One way to examine this process of mixing is to quantify, for each lipid in the simulation box, what fraction of all other lipids in the same leaflet are at some point part of their first solvation shell. In our case, this analysis shows that each lipid, on average, is in direct contact with 80% of all other lipids in the course of each of our simulations (i.e. over 1100 molecules) (*Figure 2—figure supplement 1B*). Given that at any given timepoint, a solvation shell consists of fewer than 10 lipids, this result implies extensive mixing and hence no concern in regard to the starting condition. We also examined the orientation of the protein in the bilayer, which is also an arbitrary initial condition. This analysis indicates that the simulations broadly explore orientation space, resulting in clearly defined probability distributions for both monomer and dimer (*Figure 2—figure supplement 1C*).

With sufficient lipid exchange over the time-scale of the simulations, we proceeded to analyze the shape of the lipid bilayer near the monomer and the dimer as well as other structural descriptors (*Figure 2—figure supplement 2*). From the simulated trajectories, we calculated 3D density maps reflecting the spatial distribution of both acyl chains and ester linkages in the protein vicinity (*Figure 2A,B*). The results for the monomer show that the membrane shape is deformed at the dimerization interface, thinning near the two shorter helices at the center. Elsewhere along the monomer perimeter the membrane is largely unperturbed, and its shape is nearly identical to what we observe for the dimer, confirming the simulations are probing the membrane structure reliably. Quantitative analysis of bilayer thickness, measured by the separation between the outer and inner ester layers and represented on a 2D heat map, shows that the magnitude of this thinning defect is about 8 Å relative to the bulk (*Figure 2C*), that is, nearly a quarter of the unperturbed hydrophobic thickness of this 2:1 POPE/POPG membrane. This defect is also clearly specific to the dimerization

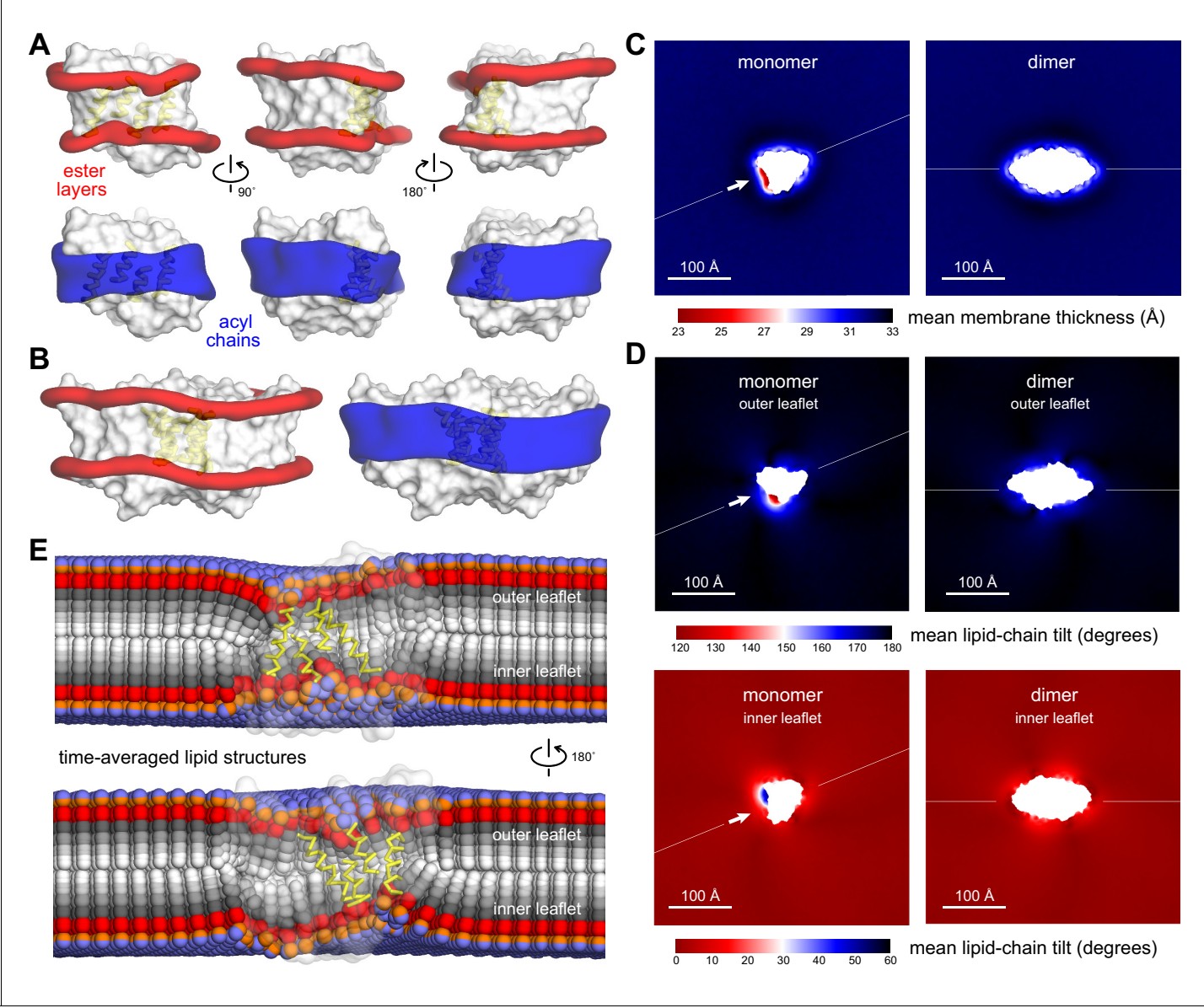

**Figure 2.** Membrane morphology around CLC-ec1 monomer and dimer from molecular dynamics simulations. Results are shown for 2:1 POPE/POPG membranes, averaging eight independent trajectories of ≈ 7.5 μs each for the monomer, and 10 trajectories of ≈ 10 μs each for the dimer (*Figure 2— source data 1*). All simulations are based on the coarse-grained MARTINI force field. (A, B) 3D density maps for the ester layers (red) or acyl chains (blue) in the vicinity (≤10 Å) of the protein (white surface), for (A) the monomer and (B) the dimer. In (A), three different views of the lipid first-shell are depicted; the four helices at dimerization interface are highlighted (yellow). (C–E) Spatially resolved grid-based analysis of different descriptors of the lipid bilayer structure. See *Figure 2—figure supplement 2* for details. (C) 2D maps of the local bilayer thickness across the simulation system. The proteins occupy the central area (white mask). Note monomer and dimer are oriented differently relative to the periodic boundaries of the simulation system. To facilitate this comparison, an axis perpendicular to the dimerization interface (white arrow) is drawn in both cases. (D) Variation in the mean lipid tilt-angle across the membrane plane, relative to the bilayer perpendicular, for both the outer and inner leaflets. (E) Time-averages of the instantaneous 3D conformation of lipid molecules residing at different positions across the membrane plane. Acyl chains (gray scale), ester linkages (red), and headgroups (orange/purple) are shown as spheres. Note perfectly isotropic dynamics, when time-averaged, results in a linear structure for the entire molecule, perpendicular to the membrane mid-plane, and with both acyl chains superposed. These structures are therefore non-physical, but they reveal the mean tilt of the lipid molecules across the membrane as well as the degree of contacts between leaflets. Data is shown for the monomer only (transparent surface), viewed from two sides. The four helices at the dimerization interface are highlighted (yellow). See *Figure 2—figure supplements 1–3*, *Figure 2—source data 1* for additional details.

The online version of this article includes the following source data and figure supplement(s) for figure 2:

**Source data 1.** CGMD simulation specifications.

**Source data 2.** Simulation files for CLC-ec1 in 2:1 POPE:POPG.

*Figure 2 continued*

**Figure supplement 1.** Simulation systems and quantification of configurational lipid and protein sampling.
**Figure supplement 2.** Schematic of different descriptors of lipid-bilayer structure.
**Figure supplement 3.** Membrane morphology around CLC-ec1 monomer and dimer from molecular dynamics simulations.

interface in the monomeric state, consistent with the 3D density maps. Smaller defects are discernable elsewhere but, as noted, they are indistinguishable if monomer and dimer are compared.

Membrane thickness deformations are sometimes conceptualized as resulting from spring-like compressions or extensions of the lipid chains (*Andersen and Koeppe, 2007*; *Brown, 2017*). In this case, however, the mean acyl chain end-to-end distance near the protein is only a fraction of 1 Å smaller than the bulk value (*Figure 2—figure supplement 3A*); this minor perturbation is also not specific to the dimerization interface, but is present at other regions. Therefore, the thinning defect that we observe does not arise due to a significant compression of the acyl chains. Instead, our simulations show that increased lipid-chain tilt is in large part what leads to the membrane thinning. This effect is clear from analysis of the orientation of the acyl chains in terms of the coarse-grained equivalent of a second-rank order parameter, which reveals a clear change at the dimerization interface (*Figure 2—figure supplement 3B*). To quantify this effect more directly, we evaluated the mean lipid-chain tilt angle across the system, relative to the membrane normal (*Figure 2D*). In the bulk, this angle averages to 0° for one leaflet and 180° for the other, as one would expect, as the lipid dynamics are isotropic. Approaching the dimerization interface, however, this angle increases gradually and is maximally deflected by 60°, in both leaflets. This drastic change in orientation can be clearly visualized in 3D by analyzing the 'average structure' of the lipid molecules residing at different positions along the membrane (*Figure 2E*). In the bulk, this average yields a linear structure, perfectly perpendicular to the membrane mid-plane, again due to the isotropy of the lipid configurational dynamics. However, inthe lipids that are closest to the dimerization interface (yellow helices), the acyl chains adopt tilted, non-bilayer configurations in order to optimally solvate the protein. Alongside this drastic change in tilt angle, we also observe that near the dimerization interface the acyl chains in one leaflet show a greater degree of inter-digitation with those in the other leaflet (*Figure 2—figure supplement 3C*), compared to the bulk or elsewhere along the protein perimeter.

In summary, our simulation data clearly shows that when the dimerization interface of CLC-ec1 is exposed to the lipid solvent, it deforms the surrounding membrane by thinning and twisting the bilayer structure (additional effects in lipid density and hydration will be discussed later below). To solvate this 'problematic' interface, C16:0/18:1 lipids must adopt non-bilayer configurations that are significantly tilted and more entangled with lipids in the opposite leaflet. Interestingly, the perturbations we observe are in all cases symmetric with respect to the bilayer midplane, consistent with the fact that the CLC-ec1 monomer consists of two topologically inverted structural repeats; this observation further underscores that it is the protein structure that dictates the morphology of the adjacent bilayer. Altogether, these results clearly indicate that optimal lipid solvation of the monomeric state in C16:0/18:1 lipids requires the membrane to adopt a high-energy conformation. Because dimerization completely eliminates this membrane defect, the cost of lipid solvation of monomeric CLC-ec1 must therefore translate into an attractive force. Although the precise magnitude of this stabilizing effect is not directly revealed by the results presented thus far, our single-molecule TIRF assays enable us to evaluate its significance experimentally. That is, if solvation of the membrane defect caused by monomeric CLC-ec1 indeed implies a dominant energetic penalty, then the dimerization equilibrium should be shifted toward the monomeric state by introducing lipids that are a 'better' solvent for this defect; this shift should be reflected in the measured free-energy of dimerization.

## Evaluating short-chain lipids as an alternative lipid solvent

Since the membrane defect induced by monomeric CLC-ec1 is constructed by hydrophobic thinning, we decided to test this hypothesis by introducing short-chain di-lauryl (DL) C12:0 lipids into the C16:0/18:1 PO lipid membranes while keeping the overall 2:1 PE/PG headgroup composition constant (*Figure 3A*). DL lipids are shorter than PO lipids by 4–6 carbons per chain, and are also fully saturated, losing the $\omega-9$ double bond in one chain. Before drawing any conclusions in regard to

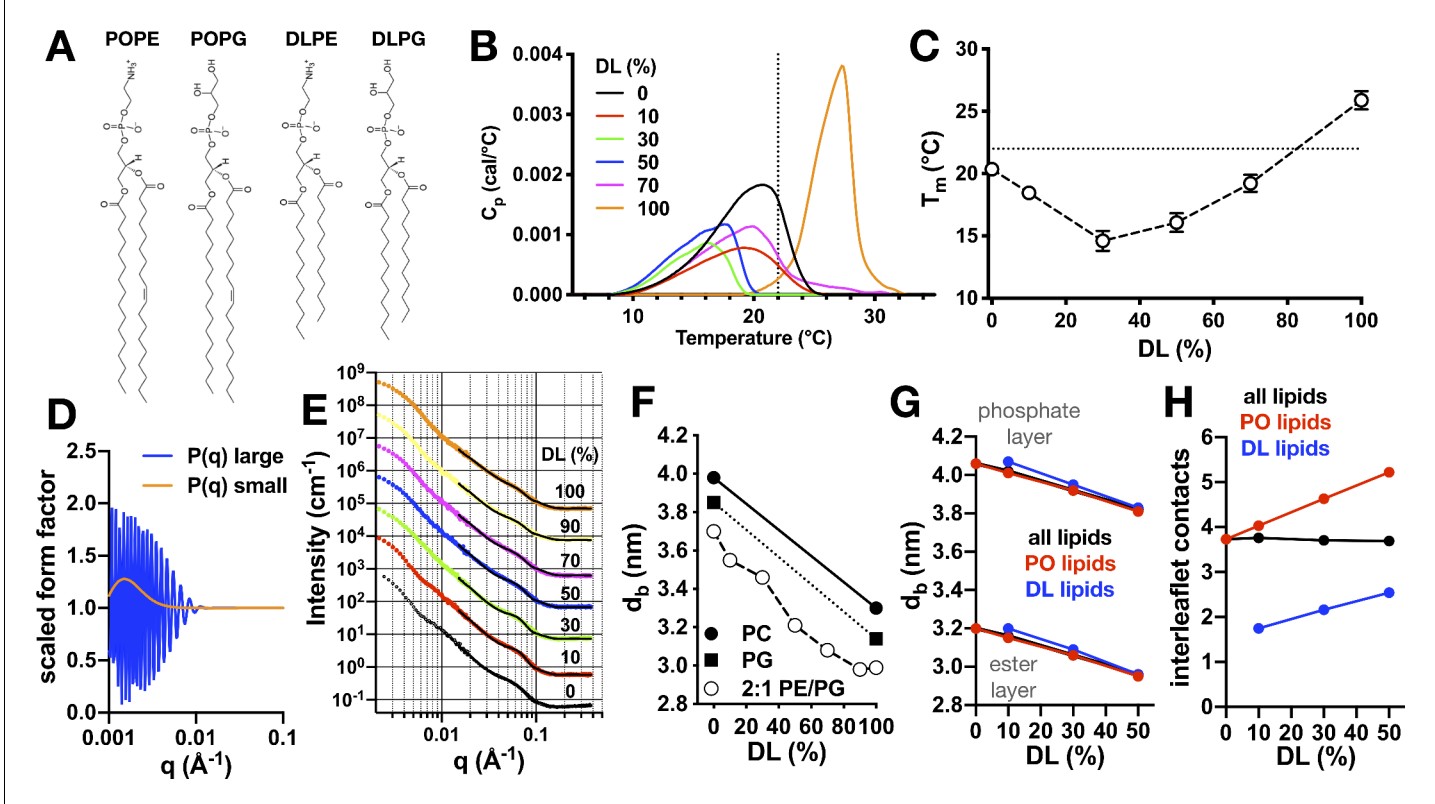

**Figure 3.** Physical properties of 2:1 PE/PG membranes with mixed PO/DL acyl chains. (**A**) Chemical structures of lipids POPE, POPG, DLPE, and DLPG. (**B**) Differential scanning calorimetry (DSC) thermograms of PO/DL mixed membranes in the multilamellar vesicle state. Dotted line marks approximate ambient room temperature (RT) of 22°C. (**C**) Peak phase transition temperature, $T_m$, as a function of DL (%). Dotted line indicates RT. (**D**) Multilamellar spherical form factors for small (orange) and large (blue) vesicle populations based on size. Both form factors are constant for q > 0.015 Å$^{-1}$. (**E**) SANS spectra as a function of DL (%) at 25°C from 100 nm extruded vesicles. Spectra are offset from the 0% condition for visualization. Solid lines represent best fit of the q > 0.015 Å$^{-1}$ regions using the multilamellar form factor model. The broad shoulder at q ≈ 0.06–0.7 is due to the presence of a small population of multi-lamellar vesicles. (**F**) Bilayer thickness ($d_B$) as a function of DL (%, white circles). Reference SANS data is shown for PC (solid circle, [*Kučerka et al., 2011*]) and PG (solid square, [*Pan et al., 2014*]). (**G**) Change in bilayer thickness as function of DL% from coarse-grained molecular dynamics simulations of pure PO/DL 2:1 PE/PG membranes, calculated as the average separation between either the phosphate layers or the ester layers. Data is shown for a calculation that considers either all lipids (black), or only PO lipids (red), or only DL lipids (blue). (**H**) Average number of contacts formed between each of the acyl chains of given lipid (either both PO and DL, black; or PO only, red; or DL only, blue) and any other acyl chain in the opposing leaflet, for the same bilayers examined in (**G**). See *Figure 3—source data 1* and *2* for additional information.

The online version of this article includes the following source data for figure 3:

**Source data 1.** Differential scanning calorimetry for mixed DL/PO 2:1 PE/PG membranes.
**Source data 2.** SANS bilayer thickness analysis for mixed DL/PO 2:1 PE/PG membranes.
**Source data 3.** Simulation files for DL/PO membranes.

the CLC-ec1 dimerization, we sought to characterize the intrinsic properties of these quaternary lipid bilayers. To do so, we first measured phase-transition thermograms for different DL/PO ratios by differential scanning calorimetry (*Figure 3B*). The mixtures show broad profiles; however, the membranes are fluid at room temperature, with the exception of the 100% DL condition. Plotting the peak $T_m$ as a function of DL shows eutectic behavior with a minimum $T_m$ at about 30% DL (*Figure 3C*). Next, we examined the structure of the DL/PO bilayers at 25°C with small-angle neutron scattering (SANS). Using a spherical, multi-lamellar liposome model to fit the scattering spectra (*Figure 3D,E*), we observe a gradual decrease in the bilayer thickness as the DL content is increased (*Figure 3F*); at 70% DL, the membrane is about 6 Å thinner than that with no DL. This change is consistent with published SANS measurements for POPC vs. DLPC (*Kučerka et al., 2011*) and POPG vs. DLPG (*Pan et al., 2014*). It also approximately matches the magnitude of the defect created by the

CLC-ec1 monomer (*Figure 2C*), indicating that the DL lipids in these mixtures might be suitable for solvating the dimerization interface.

To understand how addition of DL impacts the bilayer thickness at the molecular level, we also carried out coarse-grained molecular dynamics (CGMD) simulations for the pure PO/DL membranes. As the DL content is increased, the observed change in thickness in the simulations reproduce the experimental trend, despite the approximations inherent to the CG forcefield (*Figure 3G*). Further analysis indicates a high degree of cooperativity between the two lipid-chain types: for example, if the bilayer thickness is quantified by the distance between the two ester layers, there is virtually no difference when this distance is evaluated only for DL vs. PO lipids, at any % DL (*Figure 3G*). This observation indicates that as DL is added, their lipid headgroups remain aligned with those of the PO lipids so as to minimally perturb the degree of hydration of the headgroup layer. It is worth noting that the average number of contacts formed between the acyl chains in one leaflet and those in the other is also a conserved quantity, regardless of the PO/DL content (*Figure 3H*). For example, comparing the 0% and 10% DL membranes, we observe that a given DL chain can form only half of the interleaflet contacts seen for the PO chains in the absence of DL. However, to counter this destabilizing effect, PO lipids slightly increase the number of interactions they form across the membrane midplane and become more interdigitated; as a result, the number of chain contacts is, on average, unchanged, but this translates into a thinning of the bilayer. It appears, therefore, that an optimal degree of headgroup-layer hydration and interleaflet contacts dictates the thickness of the pure PO membrane; as DL lipids are added these PO/DL membranes adapt to preserve these two quantities, which requires them to become thinner. The significance of these conserved quantities will be discussed again further below.

## Short-chain lipids shift CLC dimerization equilibrium without effect on protein function or global membrane changes

Following these results, we investigated whether the monomer-dimer equilibrium can be influenced by addition of DLPE/PG lipids to the POPE/PG membranes, using our previously established single-molecule subunit-capture approach (*Chadda et al., 2016*; *Chadda and Robertson, 2016*). In this method, the protein is site-specifically labeled with a Cy5-maleimide fluorophore and reconstituted into lipid bilayers that are fused into large multilamellar vesicles by freeze/thaw cycles. In this state, the membrane area is sufficiently large to permit monomer and dimer populations to equilibrate according to the association constant of the reaction and the protein to lipid mole fraction, $\chi_{protein}$. The oligomeric-state distribution resulting from this equilibrium condition is quantified by fragmenting the membranes into fixed liposome compartments via extrusion, and by counting the probability distribution of subunit capture by single-molecule photobleaching analysis using TIRF microscopy. The photobleaching probability distribution follows a Poisson distribution provided one considers heterogenous compartments and multiple protein species (*Cliff et al., 2020*), and thus the population of oligomeric species can be quantified using this approach. However, it is also important to control for other factors that affect the probability distribution such as the protein labeling yield and liposome size distribution. Our site-specific labeling procedure (*Chadda et al., 2016*; *Chadda and Robertson, 2016*) provides a consistent labeling yield, $P_{Cy5,WT} = 0.663 \pm 0.005$ (mean ± sem, n = 27 independent purifications with Cy5 labeling, *Figure 4—source data 7*); thus, as long as we know the liposome size distribution, we can determine any changes of CLC-ec1 dimerization equilibrium in different lipid environments.

Using this approach, we set out to study the degree of CLC dimerization in a single mixed lipid condition in which we observe thinner membranes, namely 20% DL. Our first step was to examine the mixed DL/PO liposomes using cryo-electron microscopy imaging. 2:1 PE/PG membranes containing 20% DL and 80% PO were prepared, freeze-thawed into multi-lamellar vesicles, extruded through 400 nm filters and then imaged and analyzed to measure the size distribution directly. The liposomes and membranes appear similar to those in the 0% DL condition (i.e. 2:1 POPE/POPG), with comparable radius and fractional surface area distributions (*Figure 4A–C*). There is a significant proportion of multilamellar vesicles in both compositions, 44% for 20% DL and 25% for 100% PO samples. Next, WT-Cy5 20% DL liposomes were imaged by single-molecule TIRF microscopy. Example images and raw data for the photobleaching traces for PO and 20% DL liposomes (*Figure 4D,E*) demonstrate no changes in the quality of images obtained in the different lipid conditions. While the cryo-EM imaging indicated no significant differences in liposome size distributions, we also

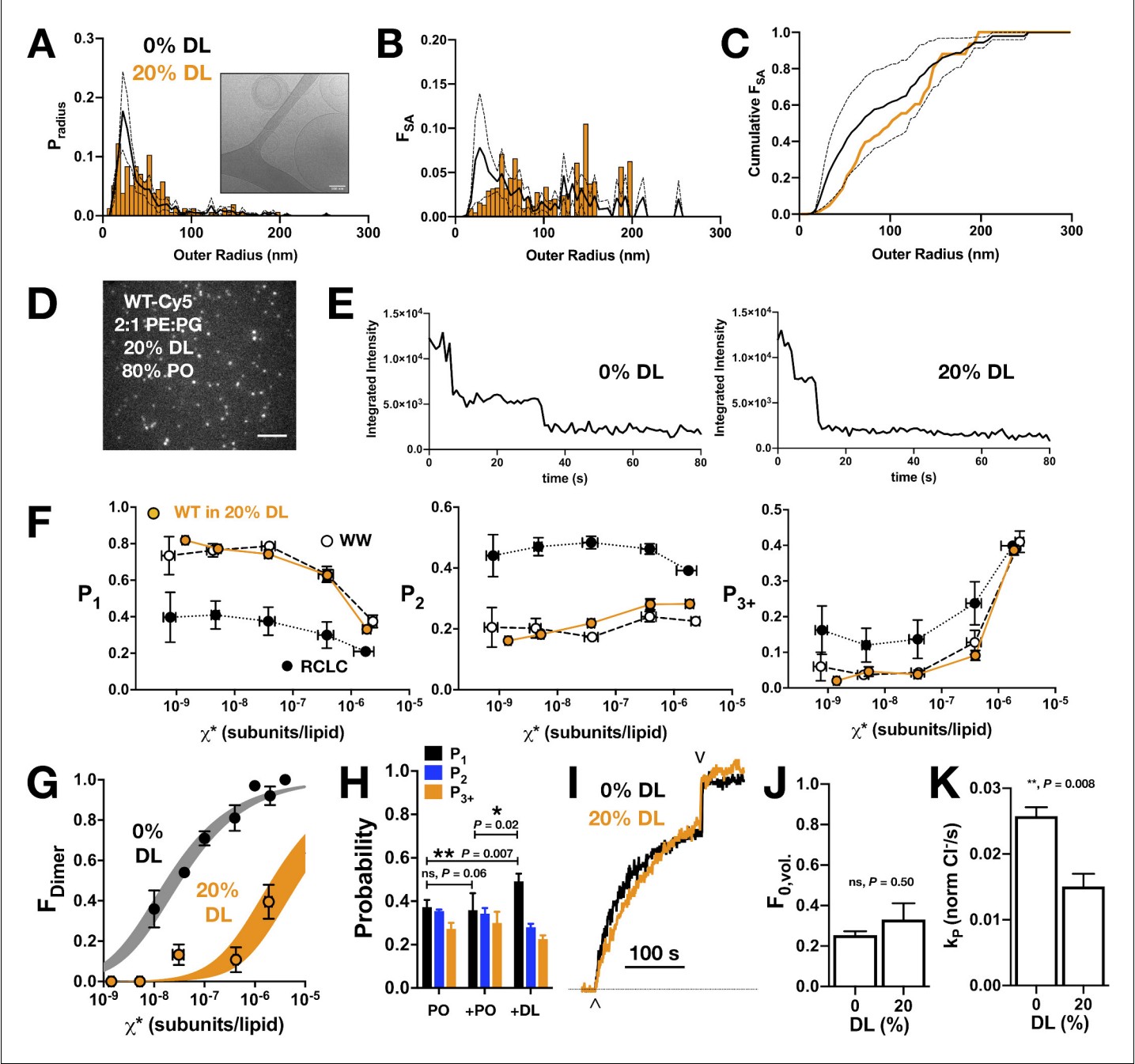

**Figure 4.** CLC-ec1 dimerization in 2:1 PE/PG membranes with 20% DL and 80% PO acyl chains. (A) Liposome size distribution of 20% DL liposomes (orange histogram), extruded through 400 nm filters and imaged by cryo-electron microscopy (inset). Black line shows mean ± standard deviation distribution of 400 nm extruded 2:1 POPE/POPG vesicles (**Cliff et al., 2020**). (B) Distribution of the fractional surface area of each liposome composition. (C) Cumulative fractional surface area distributions show that populations are not significantly different (Kolmogorov-Smirnov test, p = 0.08, D = 0.26). (D) Total internal reflection fluorescence microscopy image of 2:1 PE/PG liposomes with 20% DL containing CLC-ec1-Cy5 reconstituted at $\chi_{protein} = 1 \times 10^{-6}$ subunits/lipid ($\rho$ = 0.1 μg/mg). Scale bar represents 4 μm. (E) Representative integrated intensity photobleaching traces of WT CLC-ec1-Cy5 in 0% and 20% DL membranes. (F) Photobleaching probabilities ($P_1, P_2, P_{3+}$) of monomeric control I201W/I422W-Cy5 (WW, white circles, n = 2–3), dimeric control R230C/L249C-Cy5 (RCLC-Cy5, black circles, n = 2–3), and WT CLC-ec1-Cy5 in 20% DL (orange circles, n = 2–5), 400 nm extruded liposomes. Data represent mean ± sem for the reactive protein/lipid mole fraction $\chi^*$ (x-axis) and photobleaching probabilities (y-axis). (G) CLC-ec1 dimerization isotherm in 20% DL (orange, $K_{D,20\%DL} > (4.2 \pm 1.3) \times 10^{-6}$ subunits/lipid, $\Delta G° = -7.4 \pm 0.2$ kcal/mole, $R^2$ = 0.53, one subunit/ lipid standard state) compared to previously published 0% DL (black, $K_{D,0\% DL} = (2.5 \pm 0.4) \times 10^{-8}$ subunits/lipid, $\Delta G° = -10.9 \pm 0.1$ kcal/mole, $R^2$ = 0.92 from **Chadda et al., 2018**), resulting in $\Delta\Delta G > +3.0 \pm 0.2$ kcal/mole. (H) Fusion experiment showing the change in the photobleaching distribution of $\chi_{protein} = 2 \times 10^{-6}$ subunits/lipid CLC-ec1-Cy5 in 0% DL membranes when diluted 1:1 by fusion with 0% DL or 40% DL, for final 20% DL and $\chi_{protein} = 1 \times$

*Figure 4 continued on next page*

*Figure 4 continued*

$10^{-6}$ subunits/lipid conditions. (I) Chloride transport function for WT-Cy5 CLC-ec1 at $\chi_{protein}$ = 1 x $10^{-5}$ subunits/lipid ($\rho$ = 1 µg/mg) in 0% (black) and 20% DL (orange). Efflux is initiated by addition of valinomycin/FCCP (^) and the remaining trapped chloride is released by addition of β-OG (v). (J) Summary of the fractional volume of inactive vesicles $F_{0,vol.}$ and (K) CLC-dependent chloride efflux rate, $k_p$, over the 0% and 20% DL conditions. For the studies shown here, statistical analysis was calculated using a two-tailed unpaired parametric student's t-test on $P_1$, $F_{0,vol.}$ or $k_p$ data (*p≤0.05; **p≤0.01). See *Figure 4—figure supplement 1*, *Figure 4—source data 1–6*.

The online version of this article includes the following source data and figure supplement(s) for figure 4:

**Source data 1.** Cryo-EM radii of 20% DL 2:1 POPE/POPG.
**Source data 2.** Chloride transport of CLC-ec1-Cy5 in DL/PO proteoliposomes.
**Source data 3.** Photobleaching data for monomeric control CLC-ec1 I201W/I422W, WW-Cy5 in 20% DL, 80% PO, 2:1 PE/PG lipids.
**Source data 4.** Photobleaching data for dimeric control CLC-ec1 R230C/L249C, RCLC-Cy5 in 20% DL, 80% PO, 2:1 PE/PG lipids.
**Source data 5.** Photobleaching data for monomeric control CLC-ec1 WT-Cy5 in 20% DL, 80% PO, 2:1 PE/PG lipids.
**Source data 6.** Shift in dimer equilibrium upon fusion with DL containing vesicles.
**Source data 7.** Cy5 labeling yields.
**Figure supplement 1.** Additional analysis of CLC-ec1 assembly in membranes as a function of DL percentage in 2:1 PE/PG membranes.

examined the photobleaching probability distributions of two experimental controls: I201W/I422W, referred to as 'WW', a version of the protein with two tryptophan substitutions at the dimerization interface (*Robertson et al., 2010*) that reports the fixed monomer probability distribution; and R230C/L249C, or 'RCLC', a disulfide cross-linked constitutive dimer (*Nguitragool and Miller, 2007*) that reports the fixed dimer probability distribution (*Chadda et al., 2018*). Photobleaching analysis of these controls in 20% DL liposomes show dependencies on the protein mole fraction comparable to those observed for the 2:1 POPE/POPG composition (*Figure 4—figure supplement 1A,B*). Thus, our analyses indicate that the 20% DL, 80% PO 2:1 PE/PG liposome population is comparable to the 100% PO condition, allowing us to attribute changes in the single-molecule photobleaching distributions to specific changes in CLC dimerization.

With our quantification method benchmarked, we analyzed the photobleaching probability distribution of WT CLC-ec1 in 20% DL 2:1 PE/PG lipid bilayers and compared it to the WW and RCLC control data in the same lipid condition (*Figure 4F*). Calculation of the fraction of dimer in these protein populations, from least-squares fitting to the WW and RCLC reference distributions, shows that dimerization is significantly destabilized, that is the equilibrium is shifted toward the monomeric state (*Figure 4G*). By fitting to an equilibrium dimerization isotherm, we estimate a lower-limit of the $K_{D,Dimer}$ > 4.2 ± 1.3 x $10^{-6}$ subunits/lipid, as the reaction falls out of the dynamic range for these measurements leading to an insufficient fit of the reaction. Still, the limited reaction indicates that the 20% DL condition destabilizes dimerization by at least +3 kcal/mole. To verify that this shift reflects a new equilibrium, we also examined whether the mostly dimeric population of CLC-ec1 in 2:1 POPE/POPG lipid bilayers is driven toward the monomer state when fusing the proteo-liposomes with DL-containing membranes. *Figure 4H* shows the resultant distribution of diluting $\chi_{protein}$ = 2 x $10^{-6}$ subunits/lipid proteoliposomes 1:1 via freeze-thawed fusion with either 0% DL or 40% DL (i.e. final DL proportion is 20%). Indeed, after incubating the fused samples for 5 days at room temperature, the probability distribution showed a significant shift toward monomers, indicated by an increase in single steps, $P_1$ (*Figure 4—figure supplement 1C,D*). Therefore, alternative approaches consistently demonstrate that the short-chain DL lipid shifts the oligomeric distribution of CLC-ec1 toward the monomeric form. Finally, we examined whether CLC-ec1 remained functional in this new membrane environment. To do so, we carried out chloride efflux measurements from CLC-ec1 proteo-liposomes (*Walden et al., 2007*) and quantified the chloride transport activity as a function of DL in the membrane. The protein remained effective at transporting chloride in 20% DL (*Figure 4I*), with no difference in the fraction of inactive vesicles (*Figure 4J*), and a modest twofold decrease in chloride efflux rate (*Figure 4K*). Therefore, CLC-ec1 is significantly destabilized toward the dissociated monomeric form in 20% DL, 80% PO 2:1 PE/PG membranes, yet remains a functionally competent chloride transporter in the new lipid composition.

Next, we examined how dimerization depends on the DL/PO ratio by carrying out a titration experiment, studying the monomer-dimer population as a function of DL, from $10^{-8}$ to 80% (*Figure 5A*). Note these experiments were conducted at dilute protein densities within the membrane, with 1 subunit per million lipids ($\chi_{protein}$ = 1 x $10^{-6}$ subunits/lipid), where WT CLC-ec1 is ≈

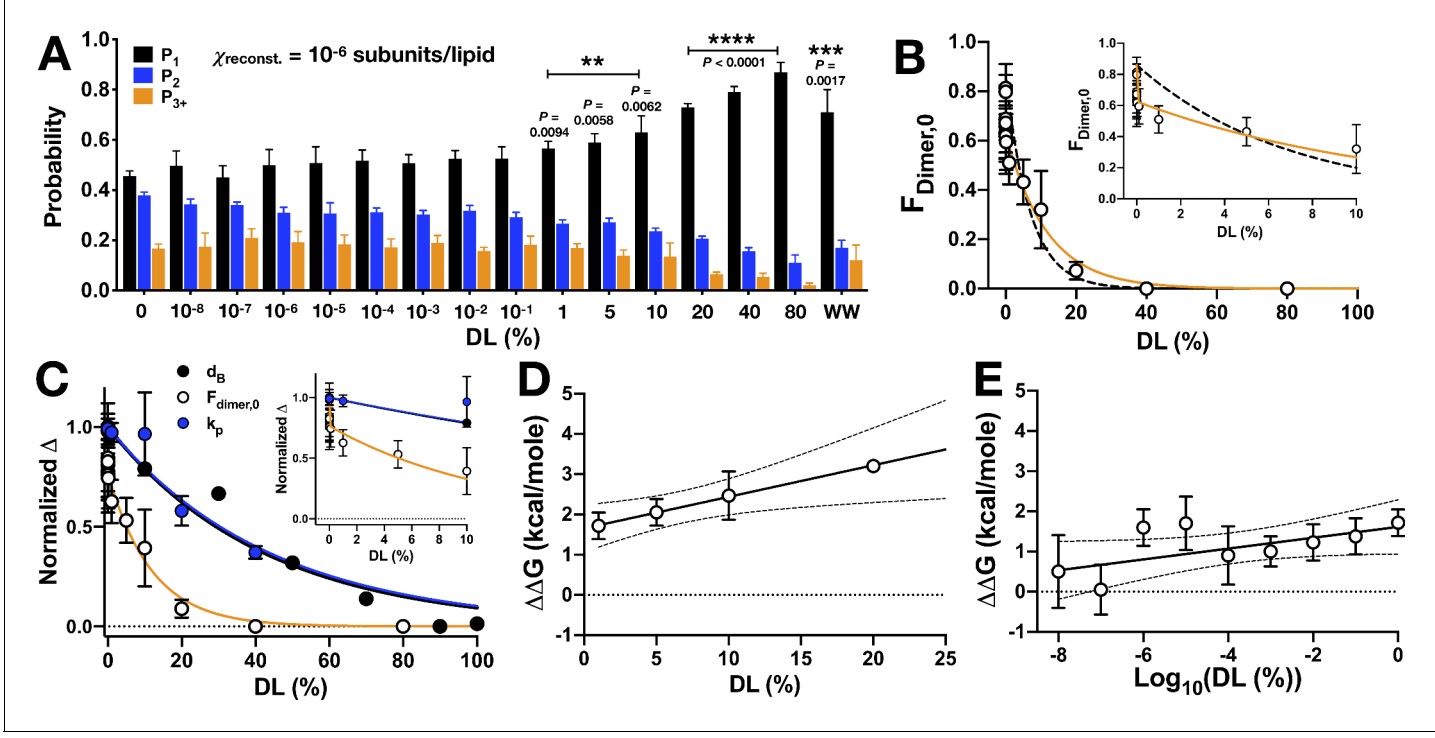

**Figure 5.** CLC-ec1 dimerization depends on DL in two phases. (A) Photobleaching probability distributions of $\chi_{protein} = 1 \times 10^{-6}$ subunits CLC-ec1-Cy5 per lipid as a function of % DL ($n_{0\% \, DL} = 9$; $n_{1E-8 \, to \, 80\% \, DL} = 3$–5; $n_{WW-Cy5, \, 0\% \, DL} = 2$ [*Chadda et al., 2016*]). Data is represented as mean ± sem. Statistical analysis calculated using a two-tailed unpaired parametric student's t-test on $P_1$ data (*p≤0.05; **p≤0.01; ***p≤0.001; ****p≤0.0001). For additional statistical information, see *Figure 5—source data 1*. (B) Fraction of dimer, $F_{dimer,0}$, calculated by least-squares fitting of WT-Cy5 distributions to WW-Cy5 and RCLC-Cy5 monomer and dimer controls. The subscript, '0', indicates that the 0% DL control distributions were used. Data was fit to a single exponential decay, $F_{Dimer,0} = F(0)e^{-\lambda(DL)}$ (black, dashed, $\lambda = 0.13 \pm 0.04$, $R^2 = 0.48$) and two-phase exponential decay, $F_{Dimer,0} = F(0)\left(F_1e^{-\lambda 1(DL)} + (1-F_1)e^{-\lambda 2(DL)}\right)$ (orange, $\lambda_1 = (1.71 \pm 2.66) \times 10^6$, $F_1 = 27.23 \pm 4.27$, $\lambda_2 = 0.09 \pm 0.03$, $R^2 = 0.64$), where F(0) = 0.86, the mean value at 0% DL. The inset shows that the two-phase exponential decay is required for fitting <1% DL data. (C) Normalized change in bilayer thickness ($\Delta d_B$, black) from SANS measurements, compared to normalized fraction of dimers ($\Delta F_{dimer,0}$, white) and chloride transport rate ($\Delta k_P$, blue) as a function of % DL. Fits are single exponential decays for $\Delta d_B$: $\lambda = 0.023 \pm 0.003$, $R^2 = 0.94$ (black), $\Delta k_P$: $\lambda = 0.025 \pm 0.005$, $R^2 = 0.59$ (blue), and a two-phase exponential decay for $\Delta F_{dimer,0}$ (orange, same fit parameters as above), with the y-intercept set to 1. Complete analysis of functional data is shown in *Figure 5—figure supplement 1*. (D) ΔΔG vs. % DL for data > 1% DL. Line represents linear regression analysis with 95% confidence intervals (slope = 0.078 ± 0.029, y-intercept = 1.65 ± 0.27, best-fit ± standard error, $R^2 = 0.38$). (E) ΔΔG vs. $Log_{10}(DL)$ for data <1% DL. Line represents linear regression analysis with 95% confidence intervals (slope = 0.135 ± 0.072, y-intercept = 1.61 ± 0.33, best-fit ± standard error, $R^2 = 0.09$). See *Figure 5— figure supplement 1*, *Figure 5—source data 1–3*.

The online version of this article includes the following source data and figure supplement(s) for figure 5:

**Source data 1.** Photobleaching titration of CLC-ec1 WT-Cy5 in 2:1 PE/PG mixed DL/PO membranes.

**Source data 2.** Testing for DL contamination during dialysis.

**Source data 3.** DL and PO concentrations, mole fraction and molality in the titrated DL/PO 2:1 POPE/POPG lipid bilayers.

**Figure supplement 1.** Chloride transport activity as a function of DL titration.

80% dimeric in 2:1 POPE/POPG. Based on our experiments of WW and RCLC controls (*Figure 4F*, *Chadda et al., 2018*), we know a dimeric population is expected to yield comparable probabilities of single and double steps ($P_1 \approx P_2$), while a monomeric population will exhibit mainly single steps ($P_1 > P_2$). The reason why a dimeric population includes a significant observation of single steps is because our experimental labeling yield is $P_{Cy5} = 0.66$, and binomial statistics predicts a nearly equal proportion of singly and doubly labeled Cy5 dimers, as demonstrated by our previous theoretical simulations (*Chadda et al., 2016*; *Chadda and Robertson, 2016*; *Cliff et al., 2020*). The raw photobleaching probability distributions show a population shift from nearly all dimers in 0% DL to all monomers at 80% DL, which resembles the distribution of WW in 2:1 POPE/POPG. Calculation of $F_{dimer}$ from these data shows that the impact of DL on dimerization follows two phases with an inflection point around 1% DL (*Figure 5B*). We also examined the dependency of CLC activity on the

presence of DL in greater detail by measuring CLC dependent chloride efflux while titrating DL in the membrane (*Figure 5—figure supplement 1*). For samples with 10% DL or less, we observed no change in chloride transport activity; by contrast, at 40% DL there is approximately a 70% reduction in transport rate. To compare how dimerization and function relate to bilayer structure, we plotted the normalized change in the bilayer thickness from the SANS data, $\Delta d_B$, with the normalized change in the dimeric population, $\Delta F_{dimer}$ and the normalized change in transport rate $\Delta k_P$ (*Figure 5C*). From this plot, we can see that > 60% of the dimerization changes occur below 10% DL, that is, before there are any major changes in the macroscopic structure of the membrane. However, the change in chloride efflux rate correlates directly with the change in bilayer thickness.

Therefore, while function appears to be impacted by global changes in membrane thickness following a simple trend, our photobleaching results demonstrate that dimerization equilibrium is coupled to the membrane in a more complex manner. Converting $F_{Dimer}$ to the change in free energy of dimerization relative to the zero DL condition, $\Delta\Delta G$, highlights the two types of molecular linkage observed. At DL >1%, the complex is destabilized by $0.8 \pm 0.3$ kcal/mole for every addition of 10% DL in the membrane (*Figure 5D*). As this corresponds to the range where we observe membrane thinning, it is reasonable to assume this change is linked to the bulk properties of the membrane. However, at DL <1%, $\Delta\Delta G$ shows a linear dependency with the logarithm of DL, with a destabilization of $0.14 \pm 0.07$ kcal/mole for every $\mathrm{Log}_{10}$ change in % DL (*Figure 5E*, *Figure 5—source data 1*). This coupling describes a microscopic process effected by DL, detectable even when DL is present in minimal amounts. It also indicates the molecular mechanism, as a linear dependency of $\Delta\Delta G$ on the logarithm of the co-solvent activity corresponds to a thermodynamic linkage model of preferential solvation, as described by *Tanford, 1969*; *Record and Anderson, 1995*; *Marsh, 1995*; *Timasheff, 2002a*.

## Preferential solvation by DL at the CLC-ec1 dimerization interface

To investigate whether preferential solvation is involved in the mechanism by which DL shifts CLC-ec1 dimerization equilibrium toward monomers, we again turned to CGMD simulations. Specifically, we carried out simulations of the CLC monomer in bilayers of DLPE/DLPG/POPE/POPG lipids with a 2:1 PE/PG ratio and a DL content of either 1%, 10%, 30%, or 50% (*Figure 6—figure supplement 1A*). Before drawing any conclusions from these simulations, we again ascertained that these complex bilayers do appropriately mix in the timescale of the trajectories, using a metric identical to that considered for the POPE/POPG simulations. Taking the 50% DL membrane as an example, we observe that by the end of the simulations, any one PO or DL lipid has been in direct contact with about 90% of all other lipid molecules in the same leaflet (*Figure 6—figure supplement 1B*), indicating near-ideal equilibration.

We then proceeded to examine the structure of these membranes using the same descriptors as those employed above. Interestingly, the thinned-membrane defect around the dimerization interface is still observed in the mixed PO/DL simulations, even for 50% DL (*Figure 6A*), and is comparable to what we find in the 2:1 POPE/POPG simulations (*Figure 2C*). That is, the defect is observed even when the 'macroscopic' thickness of the lipid bilayer is reduced; for example, for 50% DL, the thinning is about 5.5 Å. Examination of the average lipid structure across the membrane show that both PO and DL lipids solvate the totality of the protein surface and that, as in the 100% PO condition, the thinned-membrane defect results from both types of lipids becoming increasingly tilted as they approach the dimerization interface (*Figure 6B*, *Figure 6—figure supplement 2*). However, 3D density maps for the first lipid solvation shell indicate these two lipid types are not distributed identically (*Figure 6C*). At the two interfaces not involved in dimerization, the DL density signal weakens at the center of the membrane, revealing the DL chains are too short to solvate the hydrophobic span of the protein, which is better matched to PO lipids. Conversely, the density signal for PO is weaker than DL at the dimerization interface, and weaker than that seen in the 100% PO membrane, indicating PO is depleted here. This depletion, and the corresponding enrichment in DL lipids, becomes apparent in 2D projections of the percent difference between the observed lipid density ratio (DL/PO) and the expected bulk ratio (*Figure 6D*). These data show that the normalized probability of observing DL rather than PO at the dimerization interface is higher than elsewhere in the membrane, irrespective of PO/DL composition. Conversely, DL is depleted at the other two interfaces, consistent with the 3D density analysis discussed above. Quantification of the DL enrichment as

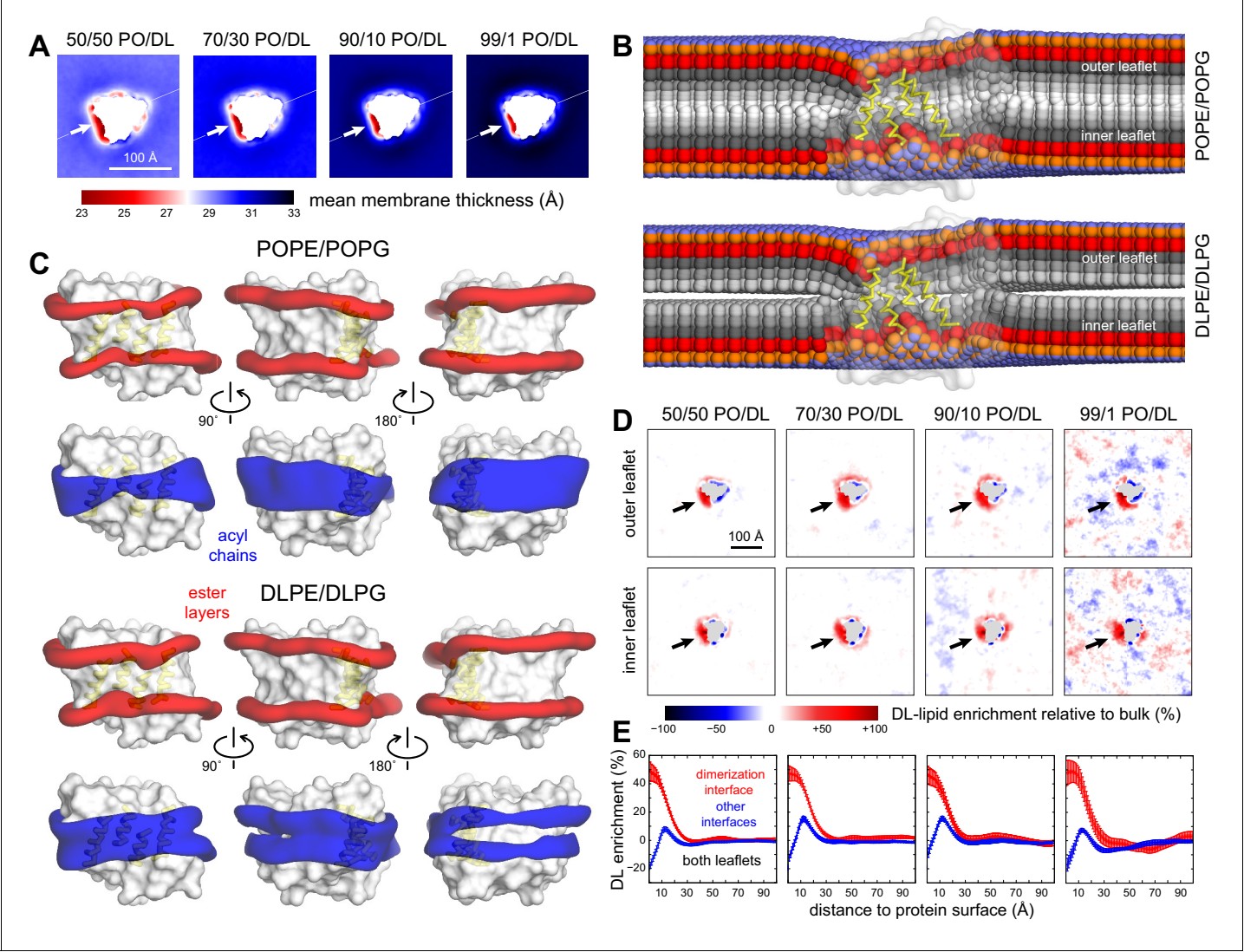

**Figure 6.** Preferential DL solvation of the CLC-ec1 dimerization interface in mixed PO/DL membranes. Data are shown for 2:1 POPE/POPG membranes with varied proportions of 2:1 DLPE:DLPG, namely 1%, 10%, 30%, and 50%. For each composition, the results presented are averages of 8 independent trajectories of 6–10 μs each (*Figure 2—source data 1*) (A) 2D maps of the bilayer thickness analogous to that shown in *Figure 2C* for a membrane with no DL. (B) For the 50% DL condition, time-averaged lipid conformations, represented identically to *Figure 2E*. PO and DL lipids are analyzed separately. (C) For the 50% DL condition, 3D density maps for the ester layers (red) or acyl chains (blue) in the vicinity (≤10 Å) of the protein (white surface), represented identically to those in *Figure 2A*. PO and DL lipids are analyzed separately. (D) Enrichment or depletion of DL lipids across the membrane, relative to the preset proportions of PO and DL lipids, quantified by the percent difference between the observed 2D lipid density ratio (DL/PO) and what would be expected for a uniform distribution and the bulk ratio. Positive values reflect enrichment while negative values reflect depletion. Each leaflet is examined separately. (E) The results shown in panel (D) are summarized by integrating the data over lipid-solvation shells of increasing width and combining the outer and inner leaflets. Independent profiles are calculated for the dimerization interface and for the other two protein-lipid interfaces. Error bars reflect the standard deviation of the data across independent trajectories.

The online version of this article includes the following source data and figure supplement(s) for figure 6:

**Source data 1.** Simulation files for CLC-ec1 in DL/PO membranes.
**Figure supplement 1.** Lipid-solvation of the CLC-ec1 monomer in CGMD simulations with PO/DL membranes.
**Figure supplement 2.** Lipid tilt around the CLC-ec1 monomer and dimer from molecular dynamics simulations.

a function of the distance from the protein surface reveals this effect extends for up to 30 Å from the dimerization interface and confirms that it is largely independent of the PO/DL ratio (*Figure 6E*).

As noted above, increasing DL content ultimately results in a change in the overall thickness of the PO/DL bilayers. Thus, it could be reasonably argued that this 'macroscopic' effect would reduce

the energetic cost of the thinned-membrane defect caused by monomeric CLC-ec1, irrespective of whether one lipid type or another is preferentially enriched, and thereby cause a shift in the dimerization equilibrium. It is important to note, however, that the enrichment effect we report is discernable at 1% DL, that is, in conditions where there is virtually no change in the global thickness of the bilayer, relative to the PO condition, both in experiment and simulation (*Figure 3F,G*). Yet, 1% DL has a profound impact on the dimerization equilibrium (*Figure 5*). Limitations in computing speed currently preclude us from verifying this effect for even smaller quantities of DL lipids with adequate statistically significance. Nevertheless, the existing results underscore that a process distinct from a change of the global properties of the membrane is dominant in this regime, which we posit is that of preferential lipid solvation.

The observation of near complete lipid mixing in our simulations implies that the enrichment of DL at the dimerization interface is neither artifactual nor transient but rather a minimum free-energy state of the lipid-solvent structure. What are the molecular factors that explain this observation, that is, what drives the preferential residence of DL over PO in this specific region of the membrane? As mentioned above, our simulation data for the pure PO/DL bilayers indicates that the collective degree of interdigitation between opposing leaflets is a conserved quantity that dictates membrane structure. This interdigitation can be quantified by metrics such as the average number of interleaflet contacts formed by the acyl-chains (*Figure 3G,H*). With this observation in mind, 2D maps of the number of interleaflet lipid contacts for the CLC-ec1 systems reveal key differences that seem to explain why DL is enriched at the dimerization interface (*Figure 7A*). In the bulk, there is no difference between the 100% and 50/50 PO/DL conditions, for example, when all lipids are averaged. The number of interleaflet contacts is again a conserved quantity, as observed for the pure bilayers. At the CLC-ec1 dimerization interface, however, this conserved quantity cannot be matched by the PO lipids; whether for 100% PO or 50/50 PO/DL, PO lipids create an excessive, clearly non-native overlap between leaflets. By contrast, when at the dimerization interface, DL chains very closely reproduce the conserved bulk values. Thus, by segregating PO lipids away from this interface, and accumulating DL lipids instead, the system minimizes the negative impact of the thinned-membrane defect created by the CLC monomer, shifting the equilibrium toward the dissociated state.

The mitigating effect of the DL lipids is also apparent from other more conventional descriptors of bilayer structure. In *Figure 7B*, for example, we quantify how the number of lipid near-neighbors varies across the membrane. The data shows that in 100% PO the CLC monomer introduces a clear lipid-density defect at the dimerization interface, particularly at the level of the headgroup and ester-linkage layers. This perturbation, in turn, leads to a marked increase in the degree of water penetration of the hydrocarbon interior of the bilayer, by almost threefold relative to the bulk-membrane values (*Figure 7C*). As noted earlier, these perturbations impact both leaflets, in a manner that reflects the internal symmetry of the CLC monomer, and very likely signify a major energetic cost. The preferential solvation of the dimerization interface by DL does not entirely eradicate these defects, but it is clear from our data that they are greatly minimized (*Figure 7B,C*). In view of these results, we can plausibly infer that in the PO/DL conditions the energetic cost of lipid solvation of the monomer is reduced, relative to the pure PO condition. This preferential solvation effect would in turn explain the shift in dimerization equilibrium observed experimentally, particularly at low DL concentrations.

## Discussion

This study demonstrates that the dimerization of a membrane protein can be driven in large part by the energetic cost of lipid solvation of the monomeric state, due to the exposure of a protein surface that deforms the membrane structure. While other factors may also contribute to the overall stability of the dimer, the significant dependency on membrane forces observed in this study implies that this oligomerization equilibrium can be regulated through variations in the chemical and physical nature of the lipid bilayer. In our studies, we observe that the monomer-dimer free-energy balance can be modulated by a mechanism of preferential solvation, that is, the enrichment of specific lipid types that are more naturally predisposed to reside in the membrane defect caused by the dissociated protomers. This is an effect that occurs with low quantities of the modulatory lipid in the membrane, before macroscopic changes in membrane thickness are apparent, and where the functionality of the protein is preserved. In contrast, when global changes in membrane thickness

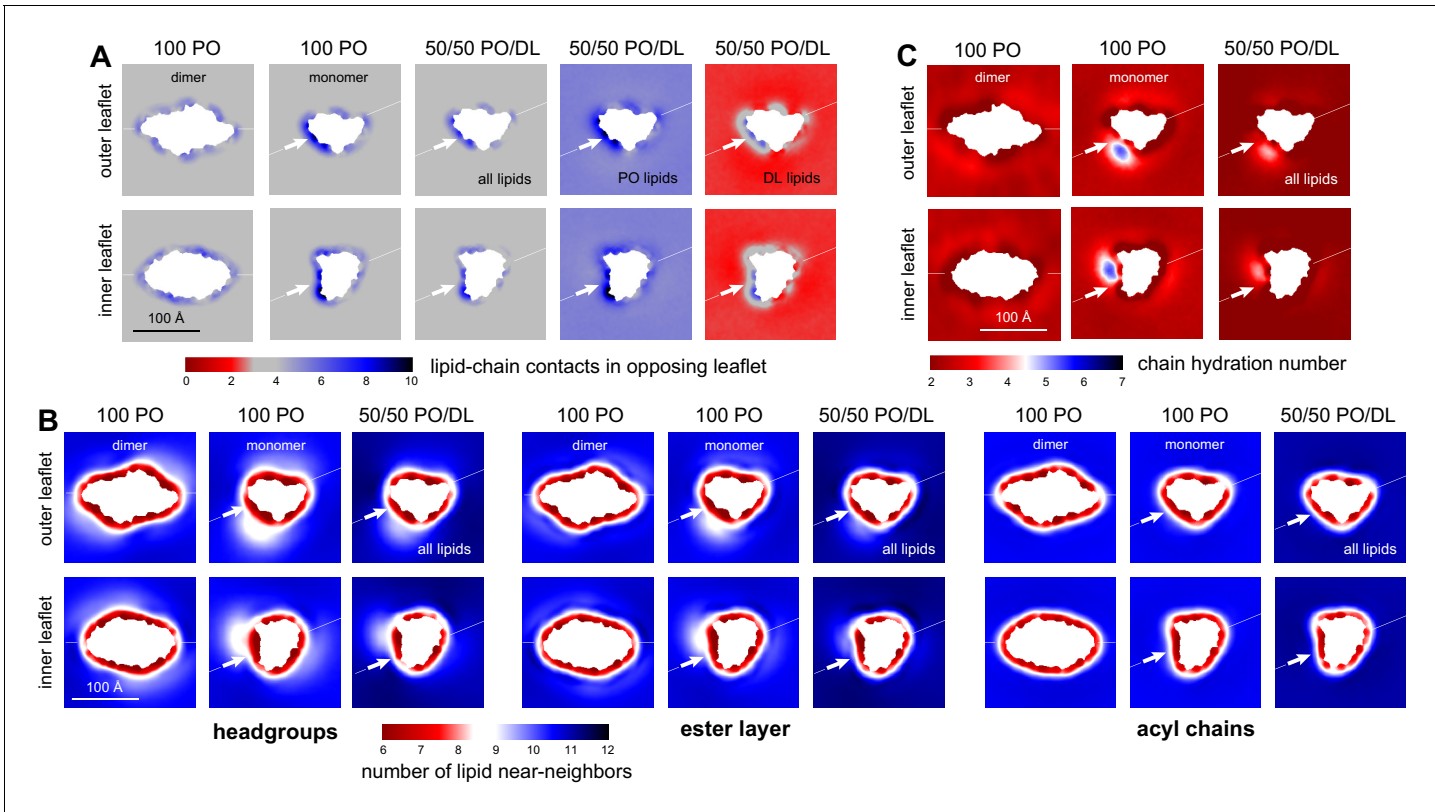

**Figure 7.** DL enrichment of membrane defect partially restores native-like properties. (**A**) Analysis of the average number of contacts formed by each of the acyl chains of a given lipid (either both PO and DL, PO only, or DL only) and any other acyl chain in the opposing leaflet. The results are mapped across the membrane plane, for either the CLC-ec1 dimer or the monomer, in either 100 PO or 50/50 PO/DL. This is the same quantity reported in *Figure 3H* for the pure PO/DL bilayers, but here it is mapped in 2D, and is calculated with the protein present. Note the number of contacts formed by DL lipids at the defect induced by monomeric CLC-ec1 approximately matches the bulk values for 100 PO or 50/50 PO/DL (regions of map in gray); by contrast, PO lipids exceed the bulk quantity. (**B**) Analysis of the 2D lipid density, at the level of either the headgroups, the ester layers, or the acyl chains, in terms of the number of lipid neighbors within 15 Å. Results are shown for each leaflet separately, for either the CLC-ec1 dimer or monomer, and in either 100 PO or 50/50 PO/DL. Note the density defect created by monomeric CLC-ec1 in the headgroup and ester layers in the 100 PO condition, and how this defect is minimized through enrichment in DL lipids. (**C**) Analysis of the extent of water penetration of the acyl-chain interior of the bilayer, for either the dimer or monomer and either the 100 PO or 50/50 PO/DL condition. Consistent with the lipid density analysis, the degree of water penetration into the bilayer interior in the 100 PO condition is much greater in the defect at the dimerization interface in monomeric CLC-ec1 than anywhere else in the membrane; this perturbation is diminished by DL enrichment.

become significant, the dimerization equilibrium has already shifted drastically, and the physical state of the membrane is observed to degrade protein function. Thus, we find that preferential solvation is a plausible contender for a mechanism of physiological regulation of membrane protein complexes in biological membranes. In the following sections, we discuss the molecular basis and physiological implications of such findings.

## Elimination of membrane defects drives CLC-ec1 dimerization

Our computational studies demonstrate that the CLC-ec1 monomer in 2:1 POPE/POPG introduces a non-native thinned defect in the surrounding membrane, due to the exposure of the shorter central helices that form the core of the dimerization interface. Experimentally, we have measured that the free energy of CLC-ec1 dimerization in 2:1 POPE/POPG lipid bilayers is $-10.9 \pm 0.1$ kcal/mole (one subunit/lipid standard state). Given the nature of the perturbations caused by the monomer, and the fact that this free-energy value can be drastically shifted by addition of minimal amounts of DLPE/DLPG lipids, we believe it is very likely that this dimerization reaction is driven largely by the energetics of the membrane, with protein-protein interactions contributing on a smaller scale. Conclusive evaluation of this hypothesis will however require further experimental and computational

investigations of the dimerization equilibrium for a range of protein constructs in different lipid bilayers and conditions, and a direct quantification of the anticipated differences in lipid solvation energetics of associated and dissociated states, in each case.

The concept that protein-induced membrane defects can translate into an effective driving force toward oligomerization presents a generalizable solution to the problem of membrane protein self-organization, while allowing for evolutionary adaptations in amino acid sequence that might be advantageous. In this perspective, association primarily depends on the overall protein architecture and the general chemical features of the protein surface. Strict conservation of specific amino acids at specific sites on the protein surface is thus not critical though not entirely inconsequential. Membrane protein complexes may thus evolve high shape-complementarity, for example to maximize the exclusion of lipids in the complexed form of the protein (*Li et al., 2013*), and thereby achieve greater stability. Indeed, previous analysis showed that CLC-ec1 exhibits high shape-complementarity, comparable to other high-affinity antigen antibody complexes (*Robertson et al., 2010*).

As noted, membrane perturbations appear to influence the association of other systems as well. Assembly of Gramicidin A peptides into functional ion-channel dimers results in a hydrophobic mismatch with the surrounding bilayer, and thus dimerization can be inhibited by increasing the global membrane thickness (*Goodall, 1971*; *Mobashery et al., 1997*). FRET measurements for reconstituted Rhodopsin have indicated the formation of higher order assemblies when the membrane is thicker or thinner than a certain range (*Botelho et al., 2006*). Computational studies have rationalized this kind of spatial organization as resulting from anisotropic defects in membrane thickness or curvature, which become mitigated upon association in specific geometries (*Mondal et al., 2014*; *Kahraman and Haselwandter, 2019*). A striking illustration of this concept is found in the inner mitochondrial membranes, where ATP synthases, each a dimeric complex, spontaneously assemble into micrometer-scale linear arrays, priming the membrane to invaginate and form cristae (*Anselmi et al., 2018*, *Blum et al., 2019*). Thus, there is a growing body of evidence that suggests that membrane-dependent forces are a key factor in the self-assembly and organization of membrane protein complexes. To our knowledge, however, this study is the first to probe how such forces can dictate the oligomerization equilibrium of a strongly-bound integral membrane protein complex, in the absence of global physical membrane changes, and without compromising its biological functionality.

## Preferential solvation vs. site-specific binding

The notion that dissociated and associated states of a membrane protein oligomer can perturb the bilayer in distinct ways, as a result of hydrophobic mismatch, implies it is conceivable that cellular mechanisms exist through which variations in lipid composition can regulate this type of equilibria. However, different mechanisms can be envisaged. The relative energetics of solvation of the dissociated and associated states would logically be altered if there is a global change in membrane thickness (*Andersen and Koeppe, 2007*). Alternatively, a different mechanism could involve that certain lipids bind to the 'problematic' protein-membrane surface, in a manner similar to conventional agonists or antagonists (*Song et al., 2013*).

Neither of these mechanisms, however, explain our experimental data. The first effect that we observe is when short-chain saturated DL lipids are added to PO bilayers at extremely low DL activities, ranging from 1 DL per $10^{10}$ PO up to 1% DL. In this regime, we determined that the bilayer thickness is identical to that of PO membranes, and we measure no change in protein function. Yet, upon increasing the amount of DL, we detect a gradual correlated increase in the monomeric proportion of CLC-ec1, and this effect appears to be linear with respect to the logarithm of DL over six orders of magnitude (*Figure 5F–H*). At first, it seems intuitive to interpret this data as a process of competitive inhibition, that is, one or more DL-specific binding sites might exist at the dimerization interface, with a lipid affinity of $K_{D,DL}$, the occupancy of which precludes dimerization. However, we can immediately see that our data do not agree with this type of linkage. This type of model would lead to complete saturation of the population of the monomeric state over a much narrower increase in % DL, at most thousand-fold, and centered at the hypothetical $K_{D,DL}$. The gradual, linear relationship of the decay of the dimeric population with the logarithm of % DL that we observe in our data, over six orders of magnitude, is simply not in agreement with a model of site-specific competitive binding.

Lipid-composition effects can be however conceptualized beyond the paradigms of bimolecular recognition or global morphological changes. If we consider lipids as solvent molecules (*Marsh, 1995*), a different type of linkage model, used to examine mixed aqueous solvent systems, explains our observations. The stability of soluble proteins, both as oligomeric assemblies or folds, is known to be dependent on the relative activities of the co-solvents present, due to preferential solvation effects (*Tanford, 1969*; *Record and Anderson, 1995*; *Schellman, 1987*; *Schellman, 2003*; *Timasheff, 2002b*; *Lee and Timasheff, 1981*). That is, one state of the protein might be 'preferentially solvated' by a given co-solvent, and so an increase in that co-solvent activity shifts the reaction equilibrium to that state. This is a form of linkage that describes how proteins can be stabilized or destabilized by salts, glycerol, sugar, or chaotropic denaturants. It does not involve specific binding, but rather non-specific affinities that lead to a linear dependency of the change in free energy with the log-activity of the co-solvent, that is, $\log(K_{eq}) \propto \log(a_{DL})$, as we observe in our experiments (*Figure 5E*).

While preferential solvation alone, that is, without bulk-membrane changes, had not been previously experimentally demonstrated to impact the formation of obligate complexes of integral membrane proteins, the notion that the features of the protein-lipid interface can dictate the spatial distribution of different lipid types in its vicinity has been previously documented. Early Monte Carlo simulations of membrane lattice models showed that consideration of differential energetics for annular lipids, and different lipid types, accounts for lateral enrichment of lipids around proteins, introducing a potential lipid-mediated driving force for aggregation in membranes (*Marcelja, 1976*; *Sperotto and Mouritsen, 1991*; *Sperotto and Mouritsen, 1993*). More recently, CGMD simulations of a wide set of membrane proteins in highly complex bilayers, it was observed that each protein induces a unique lipid solvation structure, akin to a 'lipid fingerprint' (*Corradi et al., 2018*). Similarly, a simulation study of the Gramicidin A dimer in a two-component bilayer with C16:1 and C24:1 acyl-chains, reported that the latter become underrepresented in the first solvation shell, as the C16:1 chains better match the hydrophobic thickness of the dimer (*Beaven et al., 2017*).

The experiments and simulations described in our study indeed confirm that preferential solvation effects alone can dictate the energetics of oligomerization reactions for integral membrane proteins, even those that are assumed to be obligate oligomers, where the associated form appears to be derived from evolutionarily pressures, for example CLC. That the monomer causes a thinned-membrane defect that is eliminated upon dimerization is key. We observe how at this defect the distribution of PO and DL lipids diverges from what would be expected based on their bulk ratio, and the shorter DL lipids become enriched while the PO lipids are depleted. This enrichment is specific to the dimerization interface, and therefore also specific for the monomeric state that exposes this interface. And importantly, it is observed irrespective of the PO/DL content of the bilayer, as could be expected for an effect that is dictated by the protein itself. Thus, although any deviation from the bulk-membrane homogeneity does entail a free-energy penalty, the larger gains resulting from a more optimal solvation of the exposed dimerization interface ultimately translate into a strong shift in favor of the monomeric state.

As is logical, the preferential solvation effect is ultimately superseded by more global changes in the state of the membrane; according to our SANS experiments, these changes begin to take place when the DL proportion exceeds 10%, which is the kind of change in lipid composition that has been typically evaluated in previous studies of membrane-driven organization processes. In this regime, we do observe an additional depletion of the dimer population, corresponding to the second phase of our dimerization data (*Figure 5D*), because the energetic penalty of solvating the monomer is further reduced as the membrane becomes thinner. It is possible that this change in energetic relationship reflects a change in the relative lipid energetics, however, in this high-DL range we also observe a correlated decrease in CLC-ec1 transport activity. That is, while the thinner DL/PO membranes match the exposed dimerization interface better, they also compromise the functional integrity at high DL. While we do not have direct structural information about the protein under these conditions, one plausible interpretation for this functional degradation is that the structural mechanism of the protein is somehow impaired in globally thinner membranes. Preferential solvation would thus appear to be a more viable mechanism of lipid regulation of oligomerization reactions of specific species under physiological conditions where biology strives to maintain global membrane properties.

## Physiological implications for lipid modulation of membrane proteins

Preferential solvation is a generalizable effect that could modulate any equilibrium whereby one or more protein states introduce a local morphological defect into the membrane. Besides oligomerization reactions, this effect likely defines the energetics of the intermediate conformational states that are encountered during membrane protein folding, as well as gating, transport, and signaling. Given the highly complex composition of real biological membranes, one can therefore envisage that each of these conformational states will have a different local lipid composition, optimized to stabilize the structure of the membrane in that state. The relative free-energies of the combined protein-membrane system in each state will therefore be dependent on the lipid types that are available, which the cell can alter through for example regulation of lipid synthesis and degradation pathways.

While membrane protein reactions can be severely influenced in laboratory conditions through drastic changes in the chemical and physical state of membrane, a critical point to note is that a plausible regulatory mechanism must be effective in the actual range that is physiologically viable. Cell membranes are known to undergo changes in lipid composition due to many external factors (*Marr and Ingraham, 1962*; *Sanders and Mittendorf, 2011*); yet, it is rare for a membrane to change its composition so much that its macroscopic structure is significantly altered. For instance, the membranes of *E. coli* cells grown at colder temperatures will primarily decrease chain saturation, with only minor changes to the amount of short-chain lipids. The resulting changes are presumed to maintain fluidity while maintaining an appropriate thickness of the membrane so that the majority of membrane proteins remain properly solvated and can still function optimally. Homeostatic adaptation of macroscopic membrane properties have been shown for mammalian cells also (*Levental et al., 2020*). As far as we know, there is no situation where a cell will change the overall macroscopic thickness of its membranes due to a physiological stimulus. Therefore, when contemplating possible mechanisms of physiological regulation within the membrane, and particularly with chain-length in mind, we must consider that they should be consistent with low-level changes of these types of lipids within the membrane. Our experiments show that CLC-ec1 dimerization is sensitive to the amount of the short-chain DL lipid in the membrane; even at low levels, from $10^{-8}$ to 1%, we observe a gradual and non-saturating impact, indicating that dimerization is tunable without a global change in the state of the membrane. The resulting change in the dimer population, from 80% to 50%, could certainly impart a physiological effect if it was linked to a cell signaling function. It is equally important to note that the phenomenon of preferential solvation naturally allows for this gradual tuning, as opposed to what would be expected for a process of site-specific lipid-ligand binding, which would inhibit dimerization in a switch-like manner over a much narrower range of DL-lipid concentrations. While site-specific binding mechanisms may be at play for some types of processes and specific lipid types, we anticipate that preferential solvation effects will be found to control diverse kinds of membrane protein equilibria in physiological settings.

Our examination of the impact of short-chain lipids on CLC-ec1 dimerization sheds light on a potentially ubiquitous mechanism of action by regulatory molecules within the membrane. In simulations, we observe the CLC-ec1 monomers force PO lipids to adopt non-native conformations, and many of the features of the bilayer near the dimerization interface, ranging from lipid tilt-angle to interleaflet contacts or water exclusion, are radically different from those in the bulk. When DL lipids are present, they disproportionately accumulate at this defect, spontaneously, while maintaining non-specific interactions. In doing so, DL lipids restore some of the bulk-like features to the bilayer near the dimerization interface. That is, although the defect remains, DL is a better solvent for it, and thereby stabilizes the dissociated monomeric state. A small lipid like DL could thus be considered a chemical that drives disaggregation, analogous to chaotropic denaturants stabilizing the unfolded states of proteins in aqueous solution. Many regulatory molecules in membranes are also small fatty acids; pharmacological agents like general anesthetics are small non-polar molecules as well. It is possible that these small lipoidal factors act similarly to DL in the problem examined here, and that they preferentially solvate and stabilize the membrane in states where local deformations and defects created by a protein become exposed. This may promote protein disaggregation, especially at high enough densities, and shift membrane protein equilibria to optimize activity. Altogether, our findings lead to the hypothesis that the complexity of lipid compositions found in biological cell membranes, leveraged through mechanisms such as preferential solvation, permits the cell to regulate and fine-tune the reactions of membrane proteins within – folding,

oligomerization, and conformational changes – amidst the extremely variable conditions that life faces. It will be fascinating to continue to unravel the nature of these processes through further experimental and simulation studies.

## Conclusion

This study provides fundamental insights into an ubiquitous process in membrane physiology, namely protein oligomerization. It also yields a novel perspective of the mechanism by which cells could regulate the stability of membrane protein complexes through subtle variations in the lipid composition of the bilayer. Specifically, we have posited that a principal driving force for the oligomerization of membrane proteins stems from differences in the lipid solvation energetics of the associated and dissociated states. Such differences arise when one of the states in equilibrium introduces a perturbation in the bilayer, for example due to hydrophobic mismatch, which would not be naturally observed otherwise, therefore implying a significant free-energy cost from the membrane standpoint. A driving force that originates in the energetics of lipid solvation is by definition highly sensitive to the composition of the membrane. In this regard, the perspective that emerges from this study differs from models that postulate site-specific binding or global changes in the state of the membrane. In our perspective, the lipid bilayer is a system of co-solvents that can alter their spatial distribution so as to preferentially solvate one or more of the states of any given reaction. A particular state might be therefore favored or disfavored, statistically speaking, depending on the energetics of the solvation structure that is achievable by a given co-solvent mixture. It follows that minimal changes in the lipid composition of the membrane can have a profound effect on specific oligomerization reactions, without any global morphological changes that might broadly compromise protein functionality, that is, what is expected for a physiologically realistic regulatory process.

## Materials and methods

### Coarse-grained molecular dynamics simulations

All simulations were calculated with GROMACS 5.2.1 (*Abraham et al., 2015*) using the MARTINI 2.2/ElNeDyn22 forcefield (*Wassenaar et al., 2015*). Temperature and pressure were maintained constant at 303.15 K and 1 bar, using the velocity-rescale thermostat and the Parrinello-Rahman semi-isotropic barostat, respectively. Equations of motion were integrated using the leapfrog algorithm with a time step of 20 fs. Electrostatics were treated with the reaction field method using a cutoff of 1.2 nm. To ensure statistical significance, several independent runs were performed for each system (see *Figure 2—source data 1* for further details). The simulations are based on the crystal structure of wild-type CLC-ec1 dimer deposited in the Protein Data Bank, entry 1OTS (resolution 2.51 Å) (*Dutzler et al., 2003*). Chloride ions were included at sites $S_{cen}$ and $S_{int}$, and E113 (chains A and B) and D417 (chain A) were protonated as indicated from electrostatics analysis (*Faraldo-Gómez and Roux, 2004*) In the monomer state the N-terminus was truncated up to residue 30, as this cytoplasmic helix, which domain-swaps in the dimer, is highly flexible and able to adopt alternate conformations (*Robertson et al., 2010*) The atomic structure was coarse-grained using the *martinize* tool; different mixtures of POPE, POPG, DLPE and DLPG lipids were then added around the protein and the systems solvated. The total system charge was neutralized by addition of $Na^+$ ions, and the system buffered with NaCl to a concentration of 150 mM. The preparatory stages included an 15,000-steps energy minimization using the steepest-descent method, and a 5-ns equilibration to bring the system to desired temperature and pressure. To simplify the visualization and analysis of trajectories, the protein was not permitted to rotate around the Z-axis (i.e. the membrane perpendicular) or to diffuse away from the membrane center. Note this is strictly equivalent to re-defining the laboratory frame as the molecular frame for each snapshot, and thus these restrictions have no impact on the sampling of the internal configurational space. These orientational/translational restraints were implemented with PLUMED (*Bonomi et al., 2009*); specifically, two centers-of-mass, A and B, were defined using elements of helices H/P (residues 406–409, 411, 412, 194–196, 197, 198) and the linker regions between M/N and E/F (residues 138, 143–145, 347, 348, 351–353). In the monomer simulations, harmonic potentials were used to keep both A and B on the YZ plane and equidistant from the membrane center. In the dimer, centers A and B were combined into a single center per monomer, C, and the same restraints were applied to keep the dimer on the YZ plane. The vertical drift of

the membrane was also removed prior to trajectory analysis, by re-centering each snapshot so that the midpoint of centers A and B is fixed in place. For the pure bilayer simulations, the same was accomplished by holding fixed the z-component of the membrane center. To map any given descriptor of the lipid structure onto the x-y plane, a grid consisting of square cells each with an area of 0.005 Å$^2$ was constructed. Data derived from analysis of individual lipid molecules in each simulation snapshots were mapped onto specific grid points based on the XY position of the corresponding ester beads (GL1 and GL2); specifically, data was added to all grid points contained within the van der Waals radius of the beads. The grid-point data was then averaged over the all trajectory snapshots. To ascertain which grid-points reflect statistically significant data, the frequency with which each grid point was assigned to any lipid, referred to hereafter as the occupancy number, was annotated. Grid points with less than 40% of the average occupancy were considered to be not statistically significant and excluded from graphical representations and/or global averages. Occupancy numbers were also used to quantify the enrichment of DL lipids in the mixed PO/DL systems, relative to the bulk ratio. Specifically, the percent enrichment at grid point $i$ was computed as

$$\%E_i = 100\left(\frac{\left[\frac{DL}{PO}\right]_i - \left[\frac{DL}{PO}\right]_B}{\left[\frac{DL}{PO}\right]_B}\right) \tag{1}$$

where $\rho_{DL}$ and $\rho_{PO}$ refer to the lipid occupancy number, for DLPX and POPX lipids respectively, and the subscripts $i$ and B indicates the ratio at grid point $i$ or the expected ratio in the bulk given the condition simulated, that is, if both lipid types were distributed evenly across the box. To compute the enrichment as a function of distance $d$ from the protein (or a specific interface), grid points within a mask centered at that distance and 10 Å in width were selected. The percent enrichment was then computed as

$$\%E_d = 100\left(\frac{\left[\frac{DL}{PO}\right]_M - \left[\frac{DL}{PO}\right]_B}{\left[\frac{DL}{PO}\right]_B}\right) \tag{2}$$

where the subscript M refers to the sum of the occupancy numbers over the grid points found within each mask. All grid-based analysis tools are in-house software, available for download in https://github.com/TMB-CSB/Membrane-Analysis-Tools-Gromacs (*Bernhardt and Faraldo-Gómez, 2021*), with the exception of the 3D density maps, which were calculated using the *volmap* plugin of VMD (*Humphrey et al., 1996*). For more details on the grid-based lipid metrics analysis see *Figure 2—figure supplement 2*.

## Preparation of lipids for reconstitution

Detergent solubilized lipids were prepared as described before (*Chadda et al., 2016*) with the modification that dry lipids were solubilized in 2:1 chloroform:methanol followed by two washes in 3:1 pentane:dichloromethane. This was done due to the fact that DL lipids (DLPE or 12:0 PE; 1,2-dilauroyl-sn-glycero-3-phosphoethanolamine and DLPG or 12:0 PG; 1,2-dilauroyl-sn-glycero-3-phospho-(1'-rac-glycerol) (sodium salt)), unlike PO lipids, were found to be insoluble in chloroform or pentane alone.

For a typical preparation, 4 mL of POPE, and 2 mL of POPG (25 mg/mL stocks in chloroform, Avanti Polar Lipids Inc) were combined in a glass vial (22 mm; RPI, Malvern, PA). The chloroform was evaporated under a continuous stream of 0.22 μm filtered N$_2$ gas (Ultra High Purity Nitrogen 5.0 Grade; Airgas). The dried lipid mass, was dissolved at least once in 2:1 chloroform/methanol followed by 1–2 washes in 3:1 pentane/dichloromethane and drying while rotating, leaving a thin film of lipids along the walls and the bottom of the glass vial. The lipid film, containing 150 mg total lipids (100 mg POPE +50 mg POPG) was dried under continuous stream of N$_2$, approximately 10–12 min. Next, after addition of 161.3 mg (21.5 mg/ml) CHAPS and 7.5 ml Dialysis Buffer (DB: 300 mM KCl, 20 mM citrate pH 4.5 (adjusted with NaOH)) sonication was performed leading to a translucent suspension of the CHAPS/POPE/POPG mixture. The final concentration of components was 20 mg/mL 2:1 POPE/POPG (mass ratio) and 35 mM CHAPS. DL lipids were prepared as follows: 40 mg DLPE, and 20 mg DLPG (powder, Avanti Polar Lipids Inc) were added to a glass vial. The solids were

solubilized in 2:1 chloroform/methanol and then were taken through the identical washing procedure as PO lipids until a thin, uniform lipid film resulted after drying. Next 64.5 mg CHAPS and 3 mL DB was added followed by sonication. The final concentration of components was 20 mg/mL 2:1 DLPE/DLPG (mass ratio) and CHAPS at 35 mM. Finally, the PO and DL master stocks were mixed in different ratios (volume/volume) resulting in the quaternary lipid mixtures. For instance, to prepare 1 mL of the 20% DL (w/w) lipid mixture, 0.8 mL of the PO lipid stock was mixed with 0.2 mL of the DL lipid stock and used immediately for CLC reconstitution. For reference, the conversion of % DL (w/w) to mole fraction and molality are presented in *Figure 5—source data 3*.

## Differential scanning calorimetry (DSC) experiments

The 2:1 PE/PG - PO and DL 25 mg/mL lipid stocks were solubilized in 2:1 chloroform:methanol as described above, and then mixed together to yield the following titration - 0, 10, 30, 50, 70, 90, 100 %DL. After mixing, the lipids were dried under $N_2$ gas, and solubilized in DB (10–15 mg/mL) by sonication yielding the formation of small unilamellar vesicles. These samples were freeze-thawed 7x to form multi-lamellar vesicles, which were stored at room temperature and examined by DSC days-weeks after preparation. Samples were degassed prior to measurement, and data was collected using a MicroCal VP-DSC differential scanning calorimeter. Data were collected at multiple scan rates to ensure that there was minimal influence of the scan rate on the measured melting transition. Presented data were collected on heating from 2°C to 50°C with a scan rate of 30°C/hr and were baseline corrected. Source data is provided in *Figure 3—source data 3.–source data and 1*.

## Cryo-EM measurements of liposome size distributions

Cryo-electron microscopy (EM) imaging and analysis of images was performed as described earlier (*Chadda et al., 2018*; *Cliff et al., 2020*). Briefly, liposomes were freeze-thawed seven times, and then extruded through a 400 nm nucleopore filter (GE Life Sciences) 21 times. Three µL of the undiluted sample was loaded onto a glow-discharged Lacey carbon support film (Electron Microscope Sciences), blotted, and plunged into liquid ethane using a Vitrobot System (FEI). Images were collected on a FEI Titan Krios G3 300kV Cryo-TEM microscope with a Gatan K2 Summit Direct electron detector (GATAN). Magnifications of 6500x, 33,000x, and 53,000x were used. For size determination, liposomes were manually outlined in Fiji and ImageJ (*Schindelin et al., 2012*; *Schindelin et al., 2015*) to measure the outer radii of all liposomes, including those located on the carbon. Multilamellarity was manually counted as the fraction of vesicles containing more than one bilayer. Liposome size distribution source data is provided in *Figure 4—source data 1*.

## Small-angle neutron scattering (SANS) experiments

Liposomes were prepared by drying as described previously, then sonicating the dried lipid films in reconstitution buffer prepared with 99.9% pure $D_2O$ (Cambridge Isotopes). Note, the *pD* of the buffer was measured by soaking the *pH* electrode in pure $D_2O$ for several minutes and then adjusted with NaOD for a final *pD* of 4.5 (*Krezel and Bal, 2004*). Prior to measurement, liposomes were freeze-thawed following the procedure described previously, then extruded in two steps, first through 400 nm filters and then through 100 nm nucleopore membranes.

SANS data were collected on the NGB30SANS instrument at the NIST Center for Neutron Research at the National Institute of Standards and Technology (NIST). Data were collected using a neutron wavelength ($\lambda$) 6 Å and a wavelength spread ($\Delta\lambda/\lambda$) of 0.12 with sample to detector distances of 1 m, 4 m, and 13 m. Additional data were collected using $\lambda$ = 8.4 Å with a sample to detector distance of 13 m. These instrument configurations provided access to a q-range of 0.001 Å$^{-1}$ < q < 0.04 Å$^{-1}$ where q is the scattering vector and is defined as q = $4\pi\lambda^{-1}\sin(\theta/2)$ and $\theta$ is the scattering angle. Samples were sealed in titanium cells with quartz windows and sample temperature was controlled at 25°C (±0.1°C) during data acquisition. Data were reduced to absolute intensity using the macros provided by NIST (*Kline, 2006*).

SANS data were analyzed with the multilamellar form factor in the SasView application. The data showed a broad shoulder at q $\approx$ 0.06 Å$^{-1}$ due to the presence of a mixture of unilamellar and multi-lamellar vesicles (*Scott et al., 2019*). SANS data were fit with an array distribution of N, where N is the number of lamellar shells and the reported results are for the distribution that gave the best fit to the data, defined as the minimum $\chi^2$ value. Approximately 85–90% of the vesicle population

contained a single lamella which was in good agreement with Cryo-EM experiments that confirmed the presence of ≈85% unilamellar, ≈ 10% bi-lamellar (vesicles containing two bilayers) and ≈5% multilamellar vesicles (vesicles with three or more bilayers).

Cryo-EM imaging of the liposomes also showed a bimodal distribution of vesicle sizes. The SANS analysis fixed the distribution of outer vesicle radii based on the cryo-EM results and only fit the data for q > 0.015 Å$^{-1}$ where the form factor contribution from the vesicle radii were constant (*Pencer et al., 2006*). The parameters fit during the analysis were the bilayer thickness ($d_b$), the water layer thickness ($d_w$) and the scattering length density of the bilayer (results not shown). Source data is provided in *Figure 3—source data 2*.

## Protein purification, labeling, and reconstitution

DNA constructs for CLC-ec1 C85A/H234C (WT), C85A/H234C/I201W/I422W (WW) (*Chadda et al., 2016*) and C85A/H234C/R230C/L249C (RCLC) were described previously (*Chadda et al., 2018*). Expression and purification of these CLC-ec1 variants was carried out as described earlier (*Chadda et al., 2016*). Briefly, proteins were overexpressed in BL21-AI *E. coli* competent cells and extracted into 2% n-Decyl-β-D-Maltopyranoside (DM; Anatrace, Maumee OH) containing 5 mM TCEP (Tris(2-carboxyethyl)phosphine; Soltec Bioscience, Beverly, MA). After removing cellular debris by centrifugation, the protein was affinity purified using TALON cobalt affinity resin (Clontech Laboratories, Mountain View, CA) followed by size exclusion chromatography on Superdex 200 10/30 GL size exclusion column (GE Healthcare, Little Chalfont, UK) into size exclusion buffer (SEB): 150 mM NaCl, 20 mM MOPS pH 7.0, 5 mM analytical-grade DM.

Addition of TCEP during purification ensures that the engineered cysteine at the residue H234C remains reduced and available for maleimide labeling. This can be quantitatively estimated after reacting the purified protein with Ellman's reagent (DNTB, 5,5'-Dithio-bis(2-nitrobenzoic acid); Sigma-Aldrich) as described before (*Chadda et al., 2016*). Fluorescent labeling of the protein is conducted in presence of 5X Cy5-maleimide followed by separation of unreacted dye using affinity and size-exclusion chromatography. Quantification of the Cy5 labeling yield per subunit, $P_{Cy5}$, was carried out as described previously (*Chadda et al., 2016*; *Chadda and Robertson, 2016*).

For reconstitution, Cy5-labeled protein is mixed 20 mg/mL lipids in DB + 35 mM CHAPS and then dialyzed in independent buckets to prevent the possibility of cross-contamination between different lipid compositions. Note, the effect of contamination during dialysis appears negligible in our experiments, as we quantified it by photobleaching analysis and observed a small, non-significant difference (*Figure 4—figure supplement 1E*).

## Functional measurements

Chloride transport assays from 400 nm extruded liposomes were performed as described earlier (*Walden et al., 2007*; *Chadda et al., 2016*). Functional measurements were carried out 6.4 ± 6.1 days (mean ± std, n = 2–5) after freeze/thaw and sample incubation in the dark, at room temperature. Chloride transport was quantified in two ways, by fitting the initial slope by linear regression, $k_{init.}$, or fitting the full transport trace to the following exponential association function:

$$norm.\ Cl^- = F_{0,vol.}\left(1 - e^{-k_{leak}t}\right) + \left(1 - F_{0,vol.}\right)\left(1 - e^{-k_P t}\right)$$

All data are listed in *Figure 4—source data 2*.

## Single-molecule photobleaching analysis and calculations of dimerization

Proteoliposomes samples were extruded, imaged on TIRF microscope, and the videos analyzed for counting single-molecule photobleaching steps as described earlier (*Chadda et al., 2016*; *Chadda and Robertson, 2016*; *Chadda et al., 2018*). Briefly, dialyzed proteoliposomes were freeze-thawed seven times leading to formation of large multilamellar vesicles (MLVs). The samples were stored at room temperature, in the dark, with 0.02% NaN$_3$ until extrusion and single-molecule imaging. Overall, the Cy5 labeling yield was $P_{Cy5}$ = 0.663 ± 0.005 (mean ± sem, n = 27) for wild-type CLC-ec1 samples. Imaging was carried out 3–15 days after freeze-thaw and sample incubation in the dark, at room temperature. Images were analyzed as described previously using the *imscroll* software in MATLAB (*Friedman and Gelles, 2015*).

To quantify the underlying dimerization reaction from the photobleaching data, the same methods described in were followed (*Chadda et al., 2016*; *Chadda and Robertson, 2016*; *Chadda et al., 2018*). Briefly, photobleaching probability distributions ($P_1$, $P_2$, $P_{3+}$) were determined for each construct as a function of protein density and lipid composition. The fraction of dimer in the protein population, $F_{Dimer}$, was estimated by least-squares fitting of the linear combination of the probability distributions for the monomer and dimer controls under similar conditions. Equilibrium constants were obtained by fitting the data to an equilibrium dimerization isotherm,

$$F_{Dimer} = \frac{1 + 4K_{eq}\chi^* - \sqrt{1 + 8K_{eq}\chi^*}}{4K_{eq}\chi^*} \tag{3}$$

and then converted to $\Delta G° = -RTln(K_{eq})$, standard state = 1 subunit/lipid. All data is listed in *Figure 4—source data 3–6* and *Figure 5—source data 1*.

## Acknowledgements

The Robertson lab is supported by the National Institute of General Medical Science, National Institutes of Health (R01GM120260, R21GM126476) and an equipment allocation from the Extreme Science and Engineering Discovery Environment (XSEDE; MCB200217); the Faraldo-Gómez lab is funded by the Division of Intramural Research of the National Heart, Lung and Blood Institute, NIH. SCMT is grateful for support from the U Delaware Center for Neutron Science, a cooperative agreement (70NANB15H260) with the National Center for Neutron Research at NIST, U.S. Department of Commerce. Access to the NGB30 SANS instrument was provided by the Center for High Resolution Neutron Scattering, a partnership between the National Institute of Standards and Technology and the National Science Foundation under Agreement No. DMR-1508249. The identification of any commercial products or trade names does not imply endorsement or recommendation by the National Institute of Standards and Technology. This work benefitted from the use of the SasView application, originally developed under NSF award DMR-0520547. SasView contains code developed with funding from the European Union's Horizon 2020 research and innovation program under the SINE2020 project, grant agreement No 654000. Computing resources were in part provided by the NIH Supercomputer Biowulf and XSEDE, which is supported by National Science Foundation grant number ACI-1548562. Specifically, it used the Bridges system, which is supported by NSF award number ACI-1445606, at the Pittsburgh Supercomputing Center (PSC), and the Comet system, at the San Diego Supercomputing Center (SDSC). Finally, we thank Kacey Mersch and Tim Lohman for useful discussions during the preparation of this manuscript.

## Additional information

### Competing interests

Janice L Robertson: Reviewing editor, *eLife*. José D Faraldo-Gómez: Senior editor, *eLife*. The other authors declare that no competing interests exist.

### Funding

| Funder | Grant reference number | Author |
| --- | --- | --- |
| National Institute of General Medical Sciences | R01GM120260 | Janice L Robertson |
| National Institute of General Medical Sciences | R21GM126476 | Janice L Robertson |

The funders had no role in study design, data collection and interpretation, or the decision to submit the work for publication.

### Author contributions

Rahul Chadda, Nathan Bernhardt, José D Faraldo-Gómez, Janice L Robertson, Conceptualization, Resources, Data curation, Software, Formal analysis, Supervision, Funding acquisition, Validation,

Investigation, Visualization, Methodology, Writing - original draft, Project administration, Writing - review and editing; Elizabeth G Kelley, Formal analysis, Investigation, Methodology, Writing - original draft, Writing - review and editing; Susana CM Teixeira, Kacie Griffith, Alejandro Gil-Ley, Tuğba N Öztürk, Formal analysis, Investigation, Methodology, Writing - review and editing; Lauren E Hughes, Ana Forsythe, Venkatramanan Krishnamani, Investigation, Methodology, Writing - review and editing

## Author ORCIDs

Susana CM Teixeira (ID) https://orcid.org/0000-0002-6603-7936
José D Faraldo-Gómez (ID) https://orcid.org/0000-0001-7224-7676
Janice L Robertson (ID) https://orcid.org/0000-0002-5499-9943

## Decision letter and Author response

Decision letter https://doi.org/10.7554/eLife.63288.sa1
Author response https://doi.org/10.7554/eLife.63288.sa2

---

# Additional files

## Supplementary files

• Transparent reporting form

## Data availability

All data generated or analysed during this study are included in the manuscript and supporting files. Source data files have been provided for Figures 2–6.

---

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
