## [Decision Letter]

**Acceptance summary:**

Here, the authors investigate the energetics of membrane protein dimerization using a fluorescence-based method with single-molecule resolution, guided by computational analyses. They conclude that the major influence on protein dimerization energetics is due to "preferential solvation" (localized bilayer deformation) rather than bulk hydrophobic mismatch. These general ideas have been raised in previous literature, but this is the first time they have been tested experimentally at this exquisite level of detail.

**Decision letter after peer review:**

Thank you for submitting your article "Membrane transporter dimerization driven by differential lipid solvation energetics of dissociated and associated states" for consideration by *eLife*. Your article has been reviewed by four peer reviewers, including Merritt Maduke as the Reviewing Editor and Reviewer #1, and the evaluation has been overseen by Richard Aldrich as the Senior Editor. The following individual involved in review of your submission has agreed to reveal their identity: Olaf S Andersen (Reviewer #2).

The reviewers have discussed the reviews with one another and the Reviewing Editor has drafted this decision to help you prepare a revised submission.

Summary:

In this manuscript, the authors report a careful examination of the influence of lipids on membrane-protein dimerization energetics. Using a combination of computational and experimental approaches, they examine the physical properties of mixed-lipid bilayers (containing different fractions of short-chain lipids) over a broad range of compositions, and they also examine the influence of these bilayer compositions on protein dimerization and activity. They conclude that the major influence on protein dimerization energetics is due to "preferential solvation" (localized bilayer deformation) rather than bulk hydrophobic mismatch. These ideas have been raised in previous literature, but this is the first time they have been tested experimentally at this exquisite level of detail, showing that very low mole-fractions of a "well-matching" lipid can drive the monomer-dimer equilibrium towards monomer. The extraordinary sensitivity of the method, with single-molecule resolution, is impressive. A major concern is that the authors have not done justice to describing and discussing previous studies. In addition, there are several questions concerning the computational aspects of this study. We suggest that the manuscript would benefit from substantial reorganization – emphasizing the experiments, expanding discussion of the literature, and removing, explaining and/or correcting the PMF analysis.

Essential revisions:

1. Introduction: The claim that "the physical factors controlling membrane-protein assembly in membranes are largely unknown" is an overstatement. There is a literature on membrane protein dimerization that is not quoted, including: (Lewis and Engelman, 1983), (Pearson et al., 1983), (Goforth et al., 2003) and (Rokitskaya et al., 2003). Other relevant literature should be discussed and attributed more thoroughly. See partial Reference list below. (There are many additional articles published on this topic by the labs cited.)

2. Lines 114-5: The authors should quote (Marsh, 2008), who reached a similar conclusion. See also comment relating to Lines 130-4.

3. Lines 130-4: The authors seem to assume that there is no interfacial tension between the protein's transmembrane domain and the bilayer core. This is almost certainly incorrect, cf. (Marsh, 2008).

4. Lines 137-140: This is similar to what was proposed by (Mondal et al., 2013)

5. Lines 179-180: We do not follow the logic. How do you exclude chain compression? You do address most of concerns in the following text, but this statement bothers us.

6. Paragraph beginning with Line 345: this is important, but we recommend citing similar studies done by others (Beaven et al., 2017; Sodt et al., 2017). The literature on lipid enrichment adjacent to membrane proteins is contentious, as summarized in Beaven et al.

7. The color schemes used in the figures are not color-blind friendly. This is a major problem because many of the figures cannot be interpreted if the colors cannot be seen.

8. Figure 3D, E: what are the dashed lines indicating?

9. Figure 4A,B: The "fraction of inactive vesicles" does not appear to provide a rigorous analysis of the number of vesicles without active protein, as the way it is measured involves adding detergent before the activity has plateaued. This analysis can be removed, so the focus can be on the chloride efflux rate as a measure of activity.

10. Lines 263 and 269: The statements concluding that there is no difference in CLC-ec1 structure/function for samples "with less than 20% DL" are misleading. First, the experiments measure function, not structure. Second, the only measure of function is Cl^-^ flux. It has not been determined that there are no effects on H^+^ flux and coupling stoichiometry. Finally, the functional data presented show results from 10% DL and 20% DL but nothing in between. For line 263, the statement should be rephrased to "…for samples with 10% or less DL". For line 269, a more significant change is necessary. The authors should explain clearly why they chose to perform subsequent experiments using 20% DL, given that there is a significant effect (not "no change") on Cl^-^ flux rates at 20% DL. One explanation for the ~50% decrease in activity at 20% DL is the fact that (as determined in later experiments), 20% DL yields ~100% monomer, and it is known that the monomer has ~50% the activity of the dimer. (Related to this point – what is the physical basis of this decrease in activity? This should be discussed. It is not necessarily due to a change in the overall structure, as the structure of the WW mutant is not discernably different from WT.)

11. Line 286: "…allows us to achieve a consistent labeling yield". This statement needs to be explained and justified.

12. Line 292: "slight shift towards larger liposomes". This shift seems substantial to our eyes. On what basis do the authors judge it to be "slight"?

13. Line 336: "structure of the membrane or the protein". There have been no measurements of protein structure. ("protein" should be removed.)

14. Lines 326-328 and Figure 5F: this is confusing. How does P1 ≈ P2 lead you to conclude that you have "nearly all dimers." Similarly, how does P1 > P2 lead you to conclude that you have "all monomers." We agree with the latter statement by inspection of 5C, but the logic needs to be explained.

15. Lines 322-341 and Figure 5G-I. You have not made a case for two phases. Figure 5G is misleading and does not provide evidence for two phases: even a linear relation (single exponential decay) will look "steep" between 1 and 100% if you start the plot at 10^-8^. When the same data are plotted with a linear y-axis (panel H), they seem to be well fit to a single exponential. Please report all the fitted parameters (not just the "tau,") and a measure of the goodness-of-fit. Please also use a different symbol than "tau". Given the above, we are confused by the steep rise in δ-δ G in panel I; please clarify.

16. Lines 489-491. "Yet, upon increasing the amount of DL, we detect a gradual correlated increase in the monomeric proportion of CLC-ec1, and this effect appears to be linear over six orders of magnitude in the % DL (Figure 5F-H)" This is related to point 16 and is confusing: do you mean that the amount of dimer/trimer is linear over six orders of magnitude in %DL, in which case the amount of monomer should appear exponential on a log plot? or that the free energy is linear?

17. Lines 406-8: What do you mean by "hydrophobic matching"? You equally well could argue that you are dealing with hydrophobic matching, but a more dynamic matching where the match is achieved by enriching the bilayer adjacent to the protein with the "better fitting" lipid molecules. It is unclear exactly what the difference is between the whole protein being mismatched and a piece of the protein being mismatched. From the way the manuscript is written, it appears to be that lipid rearrangement reduces the strain. But in a multicomponent bilayer this will always happen regardless of whether the whole protein is mismatched or only a piece of it (see point 19; Beaven et al). So, the distinction is not clear.

18. Line 529-533. Do you have independent evidence for the effect of membrane thickness on the structural integrity of the protein? The reduction in transport activity could be a straightforward consequence of the coupling between membrane proteins' conformational preference and bilayer thickness. The statement that DL/PO membranes compromise the "structural integrity of the protein as a whole" is not supported by any data in this manuscript.

19. We have several questions and concerns regarding the simulations.

We are skeptical that the bilayer makes a 20 kcal/mol contribution to dimerization (Page 9, line 218, end of section "The dimerization interface of CLC-ec1 introduces a structural defect into the surrounding membrane"). The details of how the thickness PMF (Figure 2I) are calculated are not clearly presented and it is not obvious how to do it.

We assume from the figures that the authors computed the variation in the thickness over the grain of their thickness discretization. This would lead to a fluctuation-derived force constant but the units would be in density. That is, small discretization binning would lead to a large variance of the thickness, the ratio of discretization area over the variance (times kT) gives the PMF force constant. This force constant would have units of kcal/mol/area (not kcal/mol as they have presented it). It is this force constant that could be in theory multiplied by the variations of thickness in Figure 2C.

We think kcal/mol/area should be the correct units for an area-elastic PMF. This is similar to the area compressibility modulus (typically done in mN/m). The area compressibility and thickness compressibility are equivalent constants related by the assumption of volume incompressibility (see Nagle and Deserno Soft Matter 2019). There are problems interpreting fluctuations in local thicknesses (see Doktorova et al., BJ 2019 v116 487 where they attempt this). In our opinion it is far better to compute the area compressibility of a protein-free bilayer using the projected area, then apply this force constant to interpret variations in thickness (average changes in thickness are robust to interpret, unlike their fluctuations).

We are wondering if the authors were unable to quantitatively link their simulation derived changes in the deformation energy to the experiments due to confusion regarding the thickness PMF. It appears to be straightforward to vary the DL percentage in the simulated bilayers, compute the change in deformation energy, then predict the relative change in dimerization probability to compare to Figure 5F. It is possible that they predicted massive changes in dimerization energy from the simulations due to an error computing the PMF. Rev 3's back-of-the-envelope estimate is much smaller and still sufficiently dramatic to be impactful, perhaps 2 kcal/mol, but without the full methodology for the PMF it cannot be certain.

As a check, the authors should explain the enhancement in the deformation on a per lipid basis. For example, if DL is favored at the detect by X% over bulk, then this site has a chemical potential that can be determined from the equilibrium constant. Given that these sites emerge when the dimer is broken, the authors can compute the change in free energy from removing or adding the site. In this manner they should be able to cross-check their mechanical PMF energy. They should also be able to predict the change in dimerization with DL % from these data.

Clarify "introduces a non-bilayer thinned defect in the surrounding membrane". Clarify "up to 10 kcal/mole" and "up to 20 kcal/mole" is this an upper limit or a best estimate?

20. Given the above, we suggest the authors consider organizing the manuscript to start with Figure 5 then Figure 4, followed by Figure 2 and 6 (and 3).

References:

Beaven, A.H., A.M. Maer, A.J. Sodt, H. Rui, R.W. Pastor, O.S. Andersen, and W. Im. 2017. Gramicidin A Channel Formation Induces Local Lipid Redistribution I: Experiment and Simulation. Biophys J. 112:1185-1197.

Ellena et al., BJ 100:1280, 2011.

Goforth, R.L., A.K. Chi, D.V. Greathouse, L.L. Providence, R.E. Koeppe, II, and O.S. Andersen. 2003. Hydrophobic coupling of lipid bilayer energetics to channel function. J. Gen. Physiol. 121:477-493.

Levental, K.R., E. Malmberg, J.L. Symons, Y.Y. Fan, R.S. Chapkin, R. Ernst, and I. Levental. 2020. Lipidomic and biophysical homeostasis of mammalian membranes counteracts dietary lipid perturbations to maintain cellular fitness. Nature communications. 11:1339.

Lewis, B.A., and D.M. Engelman. 1983. Bacteriorhodopsin remains dispersed in fluid phospholipid bilayers over a wide range of bilayer thicknesses. J. Mol. Biol. 166:203-210.

Marsh, D. 2008. Protein modulation of lipids, and vice-versa, in membranes. Biochim. Biophys. Acta. 1778:1545-1575.

Mondal, S., J.M. Johnston, H. Wang, G. Khelashvili, M. Filizola, and H. Weinstein. 2013. Membrane driven spatial organization of GPCRs. Sci Rep. 3:2909.

Pearson, L.T., S.I. Chan, B.A. Lewis, and D.M. Engelman. 1983. Pair distribution functions of bacteriorhodopsin and rhodopsin in model bilayers. Biophys. J. 43:167-174.

Rokitskaya, T.I., E.A. Kotova, and Y.A. Antonenko. 2003. Tandem gramicidin chanels cross-linked by strpetavidin. J. Gen. Physiol.:submitted for publication.

Sodt, A.J., A.H. Beaven, O.S. Andersen, W. Im, and R.W. Pastor. 2017. Gramicidin A Channel Formation Induces Local Lipid Redistribution II: A 3D Continuum Elastic Model. Biophys J. 112:1198-1213.

[Editors' note: further revisions were suggested prior to acceptance, as described below.]

Thank you for resubmitting your work entitled "Membrane transporter dimerization driven by differential lipid solvation energetics of dissociated and associated states" for further consideration by *eLife*. Your revised article has been evaluated by Richard Aldrich (Senior Editor) and a Reviewing Editor and three of the original reviewers.

This revised manuscript addresses satisfactorily our concerns about the original version. There remain just a few specific comments that need to be addressed.

1. The novelty of the study resides in the experimental results, which are first-in-kind. The organization of the manuscript, where the experimental results are presented as tests of the predictions of the molecular dynamics, may well represent the actual timeline of the study, but may obscure the importance of the experimental results. Given the present organization, we request that the manuscript include a brief but comprehensive summary of earlier studies on the lateral distribution of lipids adjacent to a membrane protein, where we note the absence of citations of the pioneering studies of Ole G. Mouritsen and colleagues.

2. One thing that is not quite clear to us from the text is whether you are proposing that the energetics are a combination of a K_D_ for lipid binding to the largest deformations around the protein plus less-specific binding as DL% increases. Both of these factors would act through the membrane deformation energy and hydrophobic mismatch. That was our understanding from reading the text but we might have misinterpreted the explanation. That is, you identify two explanations and you identify two exponentials in the fit. We can understand not wanting to definitively state the mechanism but the clarity suffers somewhat. We think this would benefit from a statement of conjecture at the end?

3. The authors conclude: "In our perspective, the lipid bilayer is a system of co-solvents that can alter their spatial distribution so as to preferentially solvate one or more of the states of any given reaction." This statement we found to be so general as to be consistent with previous treatments of hydrophobic matching (e.g. Beaven et al., which found lipids redistribute to the protein). What we found especially exciting was that the deformation is so powerful that it appears some sites appear to act like strong binding sites.

---

## [Author Response]

Essential revisions:1. Introduction: The claim that "the physical factors controlling membrane-protein assembly in membranes are largely unknown" is an overstatement. There is a literature on membrane protein dimerization that is not quoted, including: (Lewis and Engelman, 1983), (Pearson et al., 1983), (Goforth et al., 2003) and (Rokitskaya et al., 2003). Other relevant literature should be discussed and attributed more thoroughly. See partial Reference list below. (There are many additional articles published on this topic by the labs cited.)2. Lines 114-5: The authors should quote (Marsh, 2008), who reached a similar conclusion. See also comment relating to Lines 130-4.

Thank you for pointing out this oversight. We have rewritten this section in the introduction, removing the previous statement, and have provided more background and citations to previous work in this area, including these studies.

3. Lines 130-4: The authors seem to assume that there is no interfacial tension between the protein's transmembrane domain and the bilayer core. This is almost certainly incorrect, cf. (Marsh, 2008).

The hypothetical contribution of an interfacial tension is now mentioned in the revised introduction (page 4, lines 102-105). As noted, it is unclear when or if this effect will be dominant, as the interfacial tension for the acyl chains and the headgroups likely have opposite signs – as noted in Marsh, BBA 2008. A recent interesting article by Dixit and Lazaridis (JCP 2020) confirms and quantifies these as competing effects, at least for a model cylindrical geometry. In future studies it will be of interest to further evaluate these effects for more realistic geometries.

4. Lines 137-140: This is similar to what was proposed by (Mondal et al., 2013)

This study is now cited. It is important to note, however, that what we examine here is the formation of an obligate, strongly-bound membrane protein complex. The free energy of association that underlies specific complex formation of GPCRs has not been determined experimentally, to our knowledge, but it appears unlikely to approximate what is observed for CLC-ec1.

5. Lines 179-180: We do not follow the logic. How do you exclude chain compression? You do address most of concerns in the following text, but this statement bothers us.

By “chain compression”, we refer to a change in the average length of the acyl chains, which we believe has been used to conceptualize membrane thickness/curvature perturbations in molecular terms. Our simulation data, however, shows that the time-averaged end-to-end distance of the acyl chains is nearly invariant (within 1 Å) across the bilayer, including in the thinned-membrane defect caused by monomeric CLC. This defect would therefore not be accurately conceptualized as resulting from spring-like compressions of the acyl chains; what we observe instead is strong chain tilt and a greater degree of interdigitation between the two monolayers.

6. Paragraph beginning with Line 345: this is important, but we recommend citing similar studies done by others (Beaven et al., 2017; Sodt et al., 2017). The literature on lipid enrichment adjacent to membrane proteins is contentious, as summarized in Beaven et al.

These studies are now cited and discussed in some detail (page 19, lines 581-591).

7. The color schemes used in the figures are not color-blind friendly. This is a major problem because many of the figures cannot be interpreted if the colors cannot be seen.

We have changed all problematic figures as requested; in particular the color scales of the 2D maps are now blue/white/red, which is visible by the majority of color-blind individuals.

8. Figure 3D, E: what are the dashed lines indicating?

This figure has been changed and the dashed lines are now removed.

9. Figure 4A,B: The "fraction of inactive vesicles" does not appear to provide a rigorous analysis of the number of vesicles without active protein, as the way it is measured involves adding detergent before the activity has plateaued. This analysis can be removed, so the focus can be on the chloride efflux rate as a measure of activity.

We have repeated the analysis by measuring the rate as the initial slope of the transport following addition of valinomycin, and present this in Figure 4 —figure supplement 2. This method assumes constant active protein across all samples, and thus reflects changes in function. We note that while the values of the rates are about 80% of the rates from the fits to the exponential association, there is no difference in the dependency on DL. Thus, the change in transport is robust and independent of the estimation of the fitting method. In addition, in the case of the 20% DL, the plateau is visible within the transport trace and thus estimation of F0 is possible. We note, that we have measured the leak rate from the DL liposomes (Figure 4 —figure supplement 2), which allows us to constrain this component of the fit. With this, we choose to keep the F0 analysis in the updated Figure 4.

10. Lines 263 and 269: The statements concluding that there is no difference in CLC-ec1 structure/function for samples "with less than 20% DL" are misleading. First, the experiments measure function, not structure. Second, the only measure of function is Cl^-^ flux. It has not been determined that there are no effects on H^+^ flux and coupling stoichiometry. Finally, the functional data presented show results from 10% DL and 20% DL but nothing in between. For line 263, the statement should be rephrased to "…for samples with 10% or less DL". For line 269, a more significant change is necessary. The authors should explain clearly why they chose to perform subsequent experiments using 20% DL, given that there is a significant effect (not "no change") on Cl^-^ flux rates at 20% DL. One explanation for the ~50% decrease in activity at 20% DL is the fact that (as determined in later experiments), 20% DL yields ~100% monomer, and it is known that the monomer has ~50% the activity of the dimer. (Related to this point – what is the physical basis of this decrease in activity? This should be discussed. It is not necessarily due to a change in the overall structure, as the structure of the WW mutant is not discernably different from WT.)

We have made the suggested changes to the text, and revised the figures to now include a single figure on the 20% DL results that includes the functional data. With this, we expect that it is clearer that our purpose of studying the system at 20% DL, is that it is a composition where there is a significant change in the membrane thickness, and the protein retains a comparable chloride transport activity, noting that a 2-fold change in rate is minimal.

While it is an interesting hypothesis that the monomer is the cause of the 2-fold decrease in transport rate, we have evidence that shows this is not the case. A change in transport activity that depended solely on the monomeric state would mean that we would expect to see a dependency of the transport rate saturating with the dimeric population. However, Figure 5C shows that this is not what is observed. At 1 and 10% DL, we see a significant shift to monomers, yet no change in chloride transport properties. We do not yet know why there is a small change in transport function, however, we speculate that in the case of 20% DL and higher, that it is related to the thinning of the lipid bilayer, and in the case of the WW construct, the change is related to the tryptophan mutations.

11. Line 286: "…allows us to achieve a consistent labeling yield". This statement needs to be explained and justified

We have added references to our previous papers where we have described the details of site-specific labeling of C85A/H234C CLC-ec1. In addition, we report the labeling yields, P_Cy5_, for all of the WT constructs, and reference the corresponding tables. Note, these references were already in the methods section.

12. Line 292: "slight shift towards larger liposomes". This shift seems substantial to our eyes. On what basis do the authors judge it to be "slight"?

We carried out a Kolmogorov-Smirnov test indicating that there is no significant shift between the 20% and 0% DL distributions, and have therefore taken this statement out.

13. Line 336: "structure of the membrane or the protein". There have been no measurements of protein structure. ("protein" should be removed.)

Corrected.

14. Lines 326-328 and Figure 5F: this is confusing. How does P1 ≈ P2 lead you to conclude that you have "nearly all dimers." Similarly, how does P1 > P2 lead you to conclude that you have "all monomers." We agree with the latter statement by inspection of 5C, but the logic needs to be explained.

This analysis has been explained in detail in our previous studies. We include a brief description of the logic in the revised manuscript (page 12, lines 369-375), but we refer the reviewers to (Chadda et al., *eLife* 2016; Chadda and Robertson MIE 2016; Chadda et al., JGP 2018; Cliff et al., BBA-Biomembranes 2019) for further details.

15. Lines 322-341 and Figure 5G-I. You have not made a case for two phases. Figure 5G is misleading and does not provide evidence for two phases: even a linear relation (single exponential decay) will look "steep" between 1 and 100% if you start the plot at 10^-8^. When the same data are plotted with a linear y-axis (panel H), they seem to be well fit to a single exponential. Please report all the fitted parameters (not just the "tau,") and a measure of the goodness-of-fit. Please also use a different symbol than "tau". Given the above, we are confused by the steep rise in δ-δ G in panel I; please clarify.

The data has been replot in Figure 5 to provide clarification. As originally stated, there are two phases; these two phases can be clearly discerned in Figure 5B, where the data is fit by a double exponential decay, while the single exponential decay function misses the fast component completely. All fitted parameters are reported in detail and now use the symbol l.

16. Lines 489-491. "Yet, upon increasing the amount of DL, we detect a gradual correlated increase in the monomeric proportion of CLC-ec1, and this effect appears to be linear over six orders of magnitude in the % DL (Figure 5F-H)" This is related to point 16 and is confusing: do you mean that the amount of dimer/trimer is linear over six orders of magnitude in %DL, in which case the amount of monomer should appear exponential on a log plot? or that the free energy is linear?

We have now replot these graphs to show the two distinct phases, with ΔΔG following a linear relationship with % DL for conditions greater that 1% (Figure 5D), and ΔΔG following a linear relationship with Log(DL) for values less than 1% DL (Figure 5E). The latter dependency follows a model of linkage of preferential solvation as introduced by Tanford (Tanford, 1969).

17. Lines 406-8: What do you mean by "hydrophobic matching"? You equally well could argue that you are dealing with hydrophobic matching, but a more dynamic matching where the match is achieved by enriching the bilayer adjacent to the protein with the "better fitting" lipid molecules. It is unclear exactly what the difference is between the whole protein being mismatched and a piece of the protein being mismatched. From the way the manuscript is written, it appears to be that lipid rearrangement reduces the strain. But in a multicomponent bilayer this will always happen regardless of whether the whole protein is mismatched or only a piece of it (see point 19; Beaven et al). So, the distinction is not clear.

We recognize we could have been clearer on this point. By “hydrophobic matching” we meant a change in the bulk membrane thickness i.e. a global rather than local effect. One of the key conclusions of this study is that kind of global change is not required to induce strong shifts in an oligomerization equilibrium; through preferential solvation effects, marginal variations in lipid composition can have a drastic impact on this type of process, while the membrane morphology is negligibly altered. We recognize the term “hydrophobic matching” is somewhat ambiguous as it does clearly differentiate between global and local effect, so to avoid misinterpretation we have omitted it altogether from the discussion.

18. Line 529-533. Do you have independent evidence for the effect of membrane thickness on the structural integrity of the protein? The reduction in transport activity could be a straightforward consequence of the coupling between membrane proteins' conformational preference and bilayer thickness. The statement that DL/PO membranes compromise the "structural integrity of the protein as a whole" is not supported by any data in this manuscript.

We recognize we should have been clearer on this point as well. We have removed this statement from the paper, and now refer to the “compromised functional integrity” of the protein in high DL conditions.

19. We have several questions and concerns regarding the simulations.We are skeptical that the bilayer makes a 20 kcal/mol contribution to dimerization (Page 9, line 218, end of section "The dimerization interface of CLC-ec1 introduces a structural defect into the surrounding membrane"). The details of how the thickness PMF (Figure 2I) are calculated are not clearly presented and it is not obvious how to do it.We assume from the figures that the authors computed the variation in the thickness over the grain of their thickness discretization. This would lead to a fluctuation-derived force constant but the units would be in density. That is, small discretization binning would lead to a large variance of the thickness, the ratio of discretization area over the variance (times kT) gives the PMF force constant. This force constant would have units of kcal/mol/area (not kcal/mol as they have presented it). It is this force constant that could be in theory multiplied by the variations of thickness in Figure 2C.We think kcal/mol/area should be the correct units for an area-elastic PMF. This is similar to the area compressibility modulus (typically done in mN/m). The area compressibility and thickness compressibility are equivalent constants related by the assumption of volume incompressibility (see Nagle and Deserno Soft Matter 2019). There are problems interpreting fluctuations in local thicknesses (see Doktorova et al., BJ 2019 v116 487 where they attempt this). In our opinion it is far better to compute the area compressibility of a protein-free bilayer using the projected area, then apply this force constant to interpret variations in thickness (average changes in thickness are robust to interpret, unlike their fluctuations).We are wondering if the authors were unable to quantitatively link their simulation derived changes in the deformation energy to the experiments due to confusion regarding the thickness PMF. It appears to be straightforward to vary the DL percentage in the simulated bilayers, compute the change in deformation energy, then predict the relative change in dimerization probability to compare to Figure 5F. It is possible that they predicted massive changes in dimerization energy from the simulations due to an error computing the PMF. Rev 3's back-of-the-envelope estimate is much smaller and still sufficiently dramatic to be impactful, perhaps 2 kcal/mol, but without the full methodology for the PMF it cannot be certain.As a check, the authors should explain the enhancement in the deformation on a per lipid basis. For example, if DL is favored at the detect by X% over bulk, then this site has a chemical potential that can be determined from the equilibrium constant. Given that these sites emerge when the dimer is broken, the authors can compute the change in free energy from removing or adding the site. In this manner they should be able to cross-check their mechanical PMF energy. They should also be able to predict the change in dimerization with DL % from these data.Clarify "introduces a non-bilayer thinned defect in the surrounding membrane". Clarify "up to 10 kcal/mole" and "up to 20 kcal/mole" is this an upper limit or a best estimate?

We recognize that this methodology was not described in sufficient detail in the original manuscript. We also believe it would be important to have a more elaborate evaluation of its underlying assumptions, and how it compares with other theoretical approaches, e.g. those based on continuum models. Since the manuscript is as long as it is, and given that it seems uncontroversial to state the membrane perturbation caused by monomeric CLC entails some energy cost, we have decided to remove a precise quantification of this cost from the manuscript, and will discuss the methodology we propose and our results elsewhere. For the public record, however, there was no error in the information presented. The units of the PMF shown in the original version of Figure 2I were correct as shown, i.e. kcal/mol. Analysis of the trajectory data indeed leads to a free-energy density, in units of kcal/mol/area, but the PMF shown results from integration of this density over an area equivalent to the average area-per-lipid. In regard to the deformation free energies, calculated free-energy shifts derived from comparison of the 100% PO condition with data for 1-50% DL (not included in the previous version) do not, in fact, reflect “massive changes in dimerization energy”, as the reviewer speculates. The calculated values are in fact comparable in magnitude and in sign to those deduced experimentally. As mentioned, these results and the underlying methodology will be reported and discussed in detail elsewhere.

20. Given the above, we suggest the authors consider organizing the manuscript to start with Figure 5 then Figure 4, followed by Figure 2 and 6 (and 3).

We strongly believe that it is logical to first present the results from the computational analysis of 100% PO membranes, as the demonstration of the membrane defect provides the rationale for using short-chain lipids in the experiment; thus, we are keeping the original order. We have however re-organized the manuscript and figures substantially, to improve its clarity and conciseness:

– Figure 2 focuses on the thickness and lipid tilt changes in 100% PO

– Figure 4 shows all of the results pertaining to the 20% DL condition

– Figure 5 has been reorganized to show the titration data of DL, with the functional data included as a supplementary figure

– Figure 6 shows the computational findings of preferential solvation

– Figure 7 shows the lipid analysis focusing on the changes between the PO and DL solvation

[Editors' note: further revisions were suggested prior to acceptance, as described below.]

This revised manuscript addresses satisfactorily our concerns about the original version. There remain just a few specific comments that need to be addressed.1. The novelty of the study resides in the experimental results, which are first-in-kind. The organization of the manuscript, where the experimental results are presented as tests of the predictions of the molecular dynamics, may well represent the actual timeline of the study, but may obscure the importance of the experimental results. Given the present organization, we request that the manuscript include a brief but comprehensive summary of earlier studies on the lateral distribution of lipids adjacent to a membrane protein, where we note the absence of citations of the pioneering studies of Ole G. Mouritsen and colleagues.

We appreciate your recognition of the novelty of these results. We also recognize that the principle of lateral organization of lipids and lipid-mediated protein aggregation has been introduced by Marčelja in 1976 and expanded on by Mouritsen and others, and have expanded the introduction and discussion to include these references.

However, we disagree that a comprehensive summary is necessary here. Our study focused on the dimerization of CLC, and was not specifically guided by Mouritsen’s previous work, though we absolutely agree that these concepts were predicted by his work and others. Still, our study was strictly guided by a detailed analysis of a physically-based model of CLC in its surrounding membrane. As such, we disagree that presentation of this computational analysis first “may obscure the importance of the experimental results”, and ask for the consideration of the reviewers that the organization of this paper has been deliberated deeply, and we have assessed that this layout represents the most logical and objective approach to investigating the CLC reaction.

Let us consider rearranging the presentation as suggested. If one looks at the structure of the CLC dimerization interface, we would see that there are shorter helices that may perturb the membrane. We would have to guess that the membrane would be thinned, and then hypothesize that a shorter chained lipid would be a better solvent for this interface. But then, without knowing any structural information about the defect, we would be forced to take a leap in logic in deciding what lipid to choose. If we pretend that we have no quantitative information about the defect structure, we hit an illogical point in our study, having to rationalize a complete guess as to why we conducted the experiments that we did. Simply put, if one hypothesizes a defect to exist, and has the ability to analyze its structure by computational modeling, then this is the most appropriate first step in the investigation. Also, it demonstrates the advantages of quantitative analyses before jumping into the guess-work of experiments. We are confident that the way we have presented the data, computational and experimental, offers the best objective investigation of the question at hand and serves as a model as to how interdisciplinary projects should be presented.

2. One thing that is not quite clear to us from the text is whether you are proposing that the energetics are a combination of a K_D_ for lipid binding to the largest deformations around the protein plus less-specific binding as DL% increases. Both of these factors would act through the membrane deformation energy and hydrophobic mismatch. That was our understanding from reading the text but we might have misinterpreted the explanation. That is, you identify two explanations and you identify two exponentials in the fit. We can understand not wanting to definitively state the mechanism but the clarity suffers somewhat. We think this would benefit from a statement of conjecture at the end?

We have already presented an argument on page 18, lines 524-537, outlining why our data does not correspond with lipid binding. For clarification, the mechanism that we demonstrate is one in which the protein introduces a thinned defect, presumably by hydrophobic mismatch, and that DL preferentially solvates this defect leading to a linkage of the dimer equilibrium to the co-solvent activity. We have revised the paper to make an effort to clarify this whenever possible.

The discussion paragraph on page 20, lines 579-493, already outlines several possible reasons why there may be a second phase in the dimerization relationship with respect to DL composition, thus we do not think any additional conjecture is needed.

3. The authors conclude: "In our perspective, the lipid bilayer is a system of co-solvents that can alter their spatial distribution so as to preferentially solvate one or more of the states of any given reaction." This statement we found to be so general as to be consistent with previous treatments of hydrophobic matching (e.g. Beaven et al., which found lipids redistribute to the protein). What we found especially exciting was that the deformation is so powerful that it appears some sites appear to act like strong binding sites.

We agree. This is a general statement because it allows for the connection of a macroscopic model such as hydrophobic matching with a molecular model, such as preferential solvation. The wording of this sentence follows the summaries that have been used to describe the exact same phenomenon for soluble protein equilibria.

For clarification, we do not identify any lipid binding sites in this study, nor is it possible to assess from the enrichment maps alone, whether enriched regions reflect a strong binding stability. In fact, our experimental observations do not agree with strong site-specific lipid binding, as the paragraph on page 18, lines 524-537, describes. We agree that there is a preferential affinity demonstrated in the enrichment maps, but our results indicate that this is achieved by non-specific preferential solvation, which should not be grouped into the same type of affinity as a specific lipid binding “site”.